# Multiclass Learning from Contradictions

**Sauptik Dhar**
LG Electronics
Santa Clara, CA 95054
sauptik.dhar@lge.com

**Vladimir Cherkassky**
University of Minnesota
Minneapolis, MN 55455
cherk001@umn.edu

**Mohak Shah**
LG Electronics
Santa Clara, CA 95054
mohak.shah@lge.com

## Abstract

We introduce the notion of learning from contradictions, a.k.a Universum learning, for multiclass problems and propose a novel formulation for multiclass universum SVM (MU-SVM). We show that learning from contradictions (using MU-SVM) incurs lower sample complexity compared to multiclass SVM (M-SVM) by deriving the Natarajan dimension for sample complexity for PAC-learnability of MU-SVM. We also propose an analytic span bound for MU-SVM and demonstrate its utility for model selection resulting in $\sim 2 - 4\times$ faster computation times than standard resampling techniques. We empirically demonstrate the efficacy of MU-SVM on several real world datasets achieving $> 20\%$ improvement in test accuracies compared to M-SVM. Insights into the underlying behavior of MU-SVM using a histograms-of-projections method are also provided.

## 1 Introduction

Many machine learning problems in domains such as, healthcare, autonomous driving, and prognostics and health management involve learning from high-dimensional data with limited labeled samples. In such domains labeling very large quantities of data is either extremely expensive, or entirely prohibitive due to the manual effort required. Standard inductive learning methods, including data intensive deep architectures [1], may not be sufficient for such high-dimensional limited-labeled-data problems. The *learning from contradictions* paradigm (popularly known as Universum learning) has shown to be particularly effective for binary classification problems of this nature [2–11]. In this paradigm, along with the labeled training data we are also given a set of unlabeled universum samples. These universum samples belong to the same application domain as the training data, but are known not to belong to either of the two classes. The rationale behind this setting comes from the fact that even though obtaining labels is very difficult, obtaining such additional unlabeled samples is relatively easier. These unlabeled universum samples act as *contradictions* and should not be explained by the binary decision rule. However, this paradigm has been mostly limited to binary classification problems making it impractical for most real applications involving classification of more than two categories. Further, this limits incorporation of *a priori* knowledge by discarding available universum data for such applications.

Previous works such as [12, 13] have hinted on adopting an Error Correcting Output Code (ECOC) based setting such as one-vs-one (OVO) and one-vs-all (OVA), where several binary Universum-SVM [12] classifiers are combined to solve the multiclass problem. However, such studies lack a complete formalization and analysis. An alternative is the adoption of a direct approach [14] where the entire multiclass problem is solved through a single larger optimization formulation by

introducing universum learning under a probabilistic framework using a logistic loss. However, the work does not clarify as to how contradictions are captured through the proposed formulation. In this paper, we propose a formalization for multiclass learning with contradictions. Following [15] for multiclass SVM's, we introduce the new Multiclass Universum SVM (MU-SVM) formulation. The proposed MU-SVM provides a unified framework for multiclass learning under universum settings, with improved performance accuracies. The main contributions of this paper are as follows:

1. **Formulation:** We formalize the notion of universum learning for multiclass SVM (M-SVM), and propose a novel direct formulation called Multiclass Universum SVM (MU-SVM).

2. **PAC Learnability:** We derive the Natarajan dimension for the MU-SVM hypothesis class and analyze its sample complexity for PAC learnability (Theorem 2). Our analysis shows that MU-SVM incurs lower sample complexity compared to M-SVM.

3. **Useful Properties:** MU-SVM reduces to: i) standard multiclass SVM in absence of universum data and ii) binary U-SVM formulation in [16], for two-class problems (proposition 2). In addition, the proposed MU-SVM is solvable through any state-of-art M-SVM solvers (proposition 3).

4. **Model Selection:** We provide a new span definition specific to MU-SVM, and following [17] derive a leave-one-out bound for MU-SVM (Theorem 3). Under additional assumptions, a computationally efficient version of the bound is also provided (Theorem 4).

5. **Empirical validation:** Empirical results demonstrate the efficacy of the proposed formulation. We also propose a histogram-of-projections approach to analyse the results (section 4).

## 2   Multiclass SVM (M-SVM)

This section introduces multiclass learning under inductive settings and the popular Crammer and Singer's (C&S) multiclass SVM (M-SVM) formulation [15] used in such settings. Although several other multiclass SVM formulations have been proposed in literature [18–24], C&S's M-SVM is among the most widely used ones. Further, compared to the most popular multiclass formulations, C&S's M-SVM provides the smallest estimation error while ensuring a small approximation error (see [25] Table 1 for details). This makes the C&S M-SVM formulation highly desirable for limited labeled samples settings. In this paper we use the C&S's M-SVM with equal misclassification costs for balanced data as an exemplar for multiclass SVM formulations under inductive settings, and refer to it as M-SVM throughout.

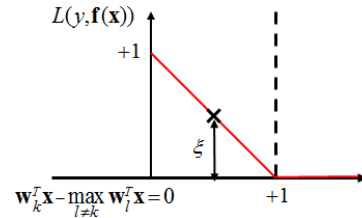

**Definition 1. (Multiclass Learning under Inductive Setting)** Given i.i.d training samples $\mathcal{T} = (\mathbf{x}_i, y_i)_{i=1}^n \sim \mathcal{D}_{\mathcal{T}}^n$, with $\mathbf{x} \in \mathcal{X} \subseteq \Re^d$ and $y \in \mathcal{Y} = \{1, \ldots, L\}$, estimate a hypothesis $h^* : \mathcal{X} \to \mathcal{Y}$ from hypothesis class $\mathcal{H}$ which minimizes,

$$\inf_{h \in \mathcal{H}} \mathbb{E}_{\mathcal{D}_{\mathcal{T}}}[\mathbb{1}_{y \neq h(\mathbf{x})}] \qquad (1)$$

Figure 1: Loss function for M-SVM with $f_k(\mathbf{x}) = \mathbf{w}_k^\top \mathbf{x}$. For the soft-margin M-SVM (3) any sample $(\mathbf{x}, y = k)$ lying inside the margin is linearly penalized using the slack variable $\xi$.

where, $\mathcal{D}_{\mathcal{T}} =$ the training distribution, $\mathbb{1}(\cdot) =$ indicator function, and $\mathbb{E}_{\mathcal{D}_{\mathcal{T}}}(\cdot) =$ expectation under training distribution.

The M-SVM, by minimizing a margin-based loss function [15], estimates $\mathbf{f} = [f_1, \ldots, f_L]$ to construct the decision rule $\hat{h}(\mathbf{x}) = \underset{l=1,\ldots,L}{\operatorname{argmax}} f_l(\mathbf{x})$. The M-SVM hypothesis class is given as,

$$\mathcal{H}_{\text{M-SVM}} = \left\{ \mathbf{x} \to \underset{l \in \mathcal{Y}}{\operatorname{argmax}} (\mathbf{w}_l^T \mathbf{x}) : \sum_{l=1}^L \|\mathbf{w}_l\|_2^2 \leq \Lambda^2; \ \mathbf{w}_k^T \mathbf{x} - \underset{l \neq k}{\operatorname{argmax}} \ \mathbf{w}_l^T \mathbf{x} \geq 1; \ \text{if } y = k \right\} \qquad (2)$$

where, $\Lambda \geq 0$ is a user-defined parameter which controls the complexity of the hypothesis class. The form in (2) is also known as the hard-margin version of M-SVM. For practical purposes we solve the

soft-margin version given below,

$$\min_{\mathbf{w}_1\dots\mathbf{w}_L,\boldsymbol{\xi}} \quad \frac{1}{2}\sum_{l=1}^{L}\|\mathbf{w}_l\|_2^2 \; + \; C\sum_{i=1}^{n}\xi_i \qquad i=1\dots n,\; l=1\dots L \tag{3}$$

$$\text{s.t.:} \quad (\mathbf{w}_{y_i}-\mathbf{w}_l)^{\top}\mathbf{x}_i \geq e_{il}-\xi_i;\; e_{il}=1-\delta_{il} \quad \delta_{il}=\begin{cases} 1; & y_i=l \\ 0; & y_i\neq l \end{cases}$$

In this formulation, the training samples falling inside the margin border ('+1') are linearly penalized using the slack variables $\xi_i \geq 0$, $i=1\dots n$ (see Fig 1), which contributes to the margin error $\sum_{i=1}^{n}\xi_i$. Eq. (3) balances between minimizing the margin error and regularization term using the user-defined parameter $C \geq 0$.

## 3 Multiclass Universum SVM (MU-SVM)

### 3.1 Multiclass U-SVM formulation

*Learning from contraditions* or Universum learning was introduced in [2] for binary classification problems to incorporate a priori knowledge about admissible data samples. For multiclass problems in addition to the labeled training data we are also given with unlabeled universum samples which are known not to belong to any of the classes in the training data. For example, if the goal of learning is to discriminate between handwritten digits (0, 1, 2,...,9), one can introduce additional 'knowledge' in the form of handwritten letters (A, B, C, ... ,Z). These examples from the universum contain certain information (e.g., handwriting styles) but they cannot be assigned to any of the classes (0 to 9). Further, the universum samples do not have the same distribution as labeled training samples. Learning under this setting can be formalized as below.

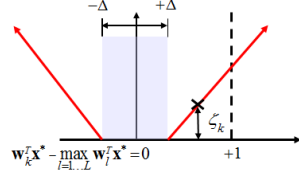

Figure 2: Loss function for universum samples $\mathbf{x}^*$ for $k^{th}$ class decision boundary $\mathbf{w}_k^{\top}\mathbf{x}^* - \max_{l=1\dots L}\mathbf{w}_l^{\top}\mathbf{x}^* = 0$. For soft-margin MU-SVM formulation (7) a sample lying outside the $\Delta$-insensitive zone is linearly penalized using the slack variable $\zeta_k$.

**Definition 2. (Multiclass Learning under Universum Setting)** Given i.i.d training samples $\mathcal{T}=(\mathbf{x}_i,y_i)_{i=1}^{n} \sim \mathcal{D}_{\mathcal{T}}^n$, with $\mathbf{x}\in\mathcal{X}\subseteq\Re^d$ and $y\in\mathcal{Y}=\{1,\dots,L\}$ and additional $m$ unlabeled universum samples $\mathcal{U}=(\mathbf{x}_{i'}^*)_{i'=1}^{m} \sim \mathcal{D}_{\mathcal{U}}$ with $\mathbf{x}^*\in\mathcal{X}_U^*\subseteq\Re^d$, estimate $h^*:\mathcal{X}\to\mathcal{Y}$ from hypothesis class $\mathcal{H}$ which, in addition to eq. (1), obtains maximum contradiction on universum samples i.e. maximizes the following probability for $\mathbf{x}^*\in\mathcal{X}_U^*$,

$$\sup_{h\in\mathcal{H}}\mathbb{P}_{\mathcal{D}_{\mathcal{U}}}[\mathbf{x}^*\notin\text{any class}] = \sup_{h\in\mathcal{H}}\mathbb{E}_{\mathcal{D}_{\mathcal{U}}}\big[\mathbb{1}_{\{\bigcap_{k\in\{1,\dots,L\}}h(\mathbf{x}^*)\neq k\}}\big] \tag{4}$$

where, $\mathcal{D}_{\mathcal{U}}$ is the universum distribution, $\mathbb{P}_{\mathcal{D}_{\mathcal{U}}}(\cdot)$ is probability under universum distribution, $\mathbb{E}_{\mathcal{D}_{\mathcal{U}}}(\cdot)$ is the expectation under universum distribution, $\mathcal{X}_U^*$ is the domain of universum data.

Learning under the universum setting has the dual goal of minimizing eq. (1) while maximizing the contradiction (in eq. (4)) on universum samples. The following proposition 1 provides guidance on how to address this for the M-SVM formulation in eq. (3).

**Proposition 1.** *For the M-SVM formulation in* (3)*, maximum contradiction on universum samples* $\mathbf{x}^*\in\mathcal{U}$ *can be achieved when,*

$$|(\mathbf{w}_k^{\top}\mathbf{x}^* - \max_{l=1\dots L}\mathbf{w}_l^{\top}\mathbf{x}^*)| = 0;\; \forall k\in\{1,\dots,L\} \tag{5}$$

That is, learning under Definition 2 using M-SVM requires the universum samples to lie on the decision boundaries of all the classes $\{1\dots L\}$. Here however, we relax this constraint (5) by requiring the universum samples to lie within a $\Delta$-insensitive zone around the decision boundaries (see Fig 2) as was done for binary scenario [16]. However, different from [16], here the $\Delta$-insensitive loss is introduced for the decision boundaries of *all* the classes i.e., $|\mathbf{w}_k^{\top}\mathbf{x}^* - \max_{l=1\dots L}\mathbf{w}_l^{\top}\mathbf{x}^*| \leq \Delta$ ;$\forall k = 1\dots L$. This reasoning motivates the new multiclass Universum-SVM (MU-SVM) formulation where the Training samples $\mathcal{T}:=(\mathbf{x}_i,y_i)_{i=1}^{n}$; $y_i\in\{1,\dots,L\}$ are penalized by standard hinge loss (similar

to M-SVM (3) and shown in Fig. 1); and the Universum samples $\mathcal{U} := (\mathbf{x}_{i'}^*)_{i'=1}^m$ are penalized by a $\Delta$-insensitive loss (see Fig. 2) for the decision functions of all the classes $\mathbf{f} = [f_1, \ldots, f_L]$.

The resulting hard-margin MU-SVM hypothesis class is given as,

$$\mathcal{H}_{\text{MU-SVM}} = \left\{ \mathbf{x} \to \underset{l=1,\ldots,L}{\operatorname{argmax}}(\mathbf{w}_l^T \mathbf{x}) : \sum_{l=1}^L \|\mathbf{w}_l\|_2^2 \leq \Lambda^2 \; ; \mathbf{w}_k^T \mathbf{x} - \underset{l \neq k}{\operatorname{argmax}} \, \mathbf{w}_l^T \mathbf{x} \geq 1; \text{ if } y = k \right.$$

$$\left. |\mathbf{w}_k^\top \mathbf{x}^* - \max_{l=1\ldots L} \mathbf{w}_l^\top \mathbf{x}^*| \leq \Delta \; ; \forall k \in \mathcal{Y} \right\} \tag{6}$$

We then relax the hard constraints on the universum samples by linearly penalizing the constraint violations through a slack variable $\zeta$ shown in Fig. 2 leading to the following soft-margin MU-SVM formulation [1],

$$\min_{\mathbf{w}_1 \ldots \mathbf{w}_L, \boldsymbol{\xi}, \boldsymbol{\zeta}} \quad \frac{1}{2} \sum_{l=1}^L \|\mathbf{w}_l\|_2^2 + C \sum_{i=1}^n \xi_i + C^* \sum_{i'=1}^m \sum_{k=1}^L \zeta_{i'k} \quad \forall i = 1 \ldots n, \quad i' = 1 \ldots m$$

$$\text{s.t.} \quad (\mathbf{w}_{y_i} - \mathbf{w}_l)^\top \mathbf{x}_i \geq e_{il} - \xi_i; \quad e_{il} = 1 - \delta_{il}, \quad l = 1 \ldots L \tag{7}$$

$$|(\mathbf{w}_k^\top \mathbf{x}_{i'}^* - \max_{l=1\ldots L} \mathbf{w}_l^\top \mathbf{x}_{i'}^*)| \leq \Delta + \zeta_{i'k}; \quad k = 1 \ldots L \quad \zeta_{i'k} \geq 0, \quad \delta_{il} = \begin{cases} 1; & y_i = l \\ 0; & y_i \neq l \end{cases}$$

Here, for the $k^{th}$ class decision boundary the universum samples $(\mathbf{x}_{i'}^*)_{i'=1}^m$ that lie outside the $\Delta$-insensitive zone are linearly penalized using the slack variables $\zeta_{i'k} \geq 0, \; i' = 1 \ldots m$. The user-defined parameters $C, C^* \geq 0$ control the trade-off between the margin size, the margin-error on training samples, and the contradictions (samples lying outside $\pm\Delta$ zone) on the universum samples. Note that for $C^* = 0$ eq. (7) reduces to the M-SVM classifier.

**Proposition 2.** *For binary classification L = 2, (7) reduces to the standard U-SVM formulation in [16] with* $\mathbf{w} = \mathbf{w}_1 - \mathbf{w}_2$ *and* $b = 0$.

### 3.2 Sample Complexity for PAC Learnability

Next we derive the sample complexity for PAC-learnability of $\mathcal{H}_{M-SVM}$ and $\mathcal{H}_{MU-SVM}$ and provide a comparative analysis. First we provide the necessary definitions,

**Definition 3. (Sample Complexity for PAC learnability [26,27])** of an algorithm $\mathcal{A} : \mathcal{X} \times \mathcal{Y} \to \mathcal{H}$ is defined as the smallest integer $n_{\mathcal{A}}(\epsilon, \delta)$ such that for any given $\epsilon, \delta > 0$ and every $n > n_{\mathcal{A}}(\epsilon, \delta)$ and distribution $\mathcal{D}$ on $\mathcal{X} \times \mathcal{Y}$ we have $\forall \, \hat{h} = \mathcal{A}((\mathbf{x}_i, y_i)_{i=1}^n)$,

$$\mathbb{P}_{(\mathbf{x}_i, y_i)_{i=1}^n \sim \mathcal{D}} \left( \mathbb{P}_D[\hat{h}(\mathbf{x}) \neq y] > \inf_{h \in \mathcal{H}} \mathbb{P}_D[h(\mathbf{x}) \neq y] + \epsilon \right) \leq \delta \tag{8}$$

The sample complexity of a hypothesis class $\mathcal{H} : n_{\mathcal{H}}(\epsilon, \delta) = \inf_{\mathcal{A}} n_{\mathcal{A}}(\epsilon, \delta)$

The sample complexity for PAC learnability depends on the size (a.k.a capacity measure) of a hypothesis class. Traditional capacity measures used in binary classification do not directly apply for multiclass problems. This has led to the research on several newer capacity measures for multiclass problems [19, 28–30]. One of the most widely researched capacity measure for multiclass SVMs is the Natarajan dimension [31] defined next,

**Definition 4. Shattering (Multiclass version)** For any hypothesis class $\mathcal{H} \subseteq \mathcal{Y}^{\mathcal{X}}$ and any $S \subseteq \mathcal{X}$, where $\mathcal{Y} = \{1, \ldots, L\}$ and $\mathcal{X} :=$ training data domain, we say $\mathcal{H}$ shatters $S$ if $\exists f_1, f_2 : S \to \mathcal{Y}$ with $\forall \mathbf{x} \in S, f_1(\mathbf{x}) \neq f_2(\mathbf{x})$ and for every $T \subseteq S$ there is a $g \in \mathcal{H}$ which satisfies,

$$\forall \mathbf{x} \in T, \; g(\mathbf{x}) = f_1(\mathbf{x}), \text{ and } \forall \mathbf{x} \in S - T, \; g(\mathbf{x}) = f_2(\mathbf{x}). \tag{9}$$

**Natarajan Dimension** $d_{\mathcal{N}}(\mathcal{H})$ is the maximal cardinality of a set that is shattered by $\mathcal{H}$.

An advantage of Natarajan Dimension is that it provides a natural extension to the fundamental learning theorem for multi-class problems (see [26, 32]) discussed next.

**Theorem 1.** *(Fundamental Learning Theorem) There exist absolute constants $C_1, C_2 > 0$ such that any hypothesis class $\mathcal{H}$ of functions from $\mathcal{X} \to \mathcal{Y}$ is PAC-learnable with sample complexity*

$$C_1 \frac{d_\mathcal{N}(\mathcal{H}) + log(1/\delta)}{\epsilon^2} \leq n_\mathcal{H}(\epsilon, \delta) \leq C_2 \frac{d_\mathcal{N}(\mathcal{H}) log(|\mathcal{Y}|) + log(1/\delta)}{\epsilon^2} \tag{10}$$

Theorem 1 shows $n_\mathcal{H}(\epsilon, \delta) = O(\frac{d_\mathcal{N}(\mathcal{H}) log(|\mathcal{Y}|) + log(1/\delta)}{\epsilon^2})$. Hence, for low sample complexity it is desirable for hypothesis classes to have smaller $d_\mathcal{N}(\mathcal{H})$. With these definitions in place, we prove a new Theorem 2 to characterize $d_\mathcal{N}(\mathcal{H})$ for $\mathcal{H}_{M-SVM}$ and $\mathcal{H}_{MU-SVM}$ as shown below,

**Theorem 2.** *The Natarajan dimension for $\mathcal{H}_{M-SVM}$ and $\mathcal{H}_{MU-SVM}$ has the form $d_\mathcal{N}(\mathcal{H}) = O(\vartheta log(\vartheta))$. Assuming $||\mathbf{x}||_2^2 \leq R^2$ ; $\forall \mathbf{x} \in \mathcal{X} \subseteq \Re^d$ gives,*

$$\mathcal{H}_{M-SVM}: \qquad \vartheta = min\, (dL + 1, 2R^2\Lambda^2) \tag{11}$$
$$\mathcal{H}_{MU-SVM}: \qquad \vartheta = min\, (dL + 1, \kappa) \tag{12}$$

*where,*

$$d = data\ dimension, \qquad L = total\ no.\ of\ classes$$

$$\kappa \leq \min_{\gamma \in \{\gamma \geq 0\, ;\, G(\gamma) \geq 0\}} \left[ F(\gamma)R^2 + \frac{\sqrt{G(\gamma)}}{2} \right] \tag{13}$$

$$F(\gamma) = \Lambda^2 + \gamma \frac{mL(L-1)}{2}\Delta^2 \qquad G(\gamma) = [2F(\gamma)R^2]^2 - 4\gamma F(\gamma)\, trace[H(\gamma)] \tag{14}$$

$$H(\gamma) = (I + \gamma \mathbf{V}\mathbf{V}^T)^{-1}(\mathbf{V}\mathbf{Z}^T\mathbf{Z}\mathbf{V}^T) \tag{15}$$

*also, the transformations $\mathbf{Z}, \mathbf{V}$ are obtained as,*

(**T1**) **Given***: For a maximally shattered set $S = \{\mathbf{x}_1, \ldots, \mathbf{x}_{d_\mathcal{N}}\}$ using the functions $f_1(\mathbf{x}), f_2(\mathbf{x})$ (see definition (4))*
**Define***: a mapping $\phi : \Re^d \to \Re^{dL}$ as ,*

$$\mathbf{z} = \phi(\mathbf{x}) = \begin{cases} (\mathbf{0}_{d \times 1}, \ldots, \underset{f_1(\mathbf{x})=l}{\mathbf{x}}, \ldots \underset{f_2(\mathbf{x})=k}{-\mathbf{x}}, \ldots, \mathbf{0}_{d \times 1})_{dL \times 1}; \ \forall \mathbf{x} \in T \subseteq S \\ (\mathbf{0}_{d \times 1}, \ldots, \underset{f_1(\mathbf{x})=l}{-\mathbf{x}}, \ldots \underset{f_2(\mathbf{x})=k}{\mathbf{x}}, \ldots, \mathbf{0}_{d \times 1})_{dL \times 1}; \ \forall \mathbf{x} \in S - T \end{cases}$$

**Obtain**: $\qquad \mathbf{Z} = \begin{bmatrix} (\mathbf{z}_1)^T \\ \vdots \\ (\mathbf{z}_{d_\mathcal{N}})^T \end{bmatrix}.$

*Basically, the transformation $\phi$ maps a sample $\mathbf{x} \in \Re^d$ from the shattered set $S$ to a $dL$ - dimension vector $\mathbf{z}$; where for any $\mathbf{x} \in T$ with $f_1(\mathbf{x}) = l$ and $f_2(\mathbf{x}) = k$; we copy the $\mathbf{x}$ vector onto $l(d-1) + 1 \ldots ld$-th position and $-\mathbf{x}$ vector onto $k(d-1) + 1 \ldots kd$-th position of $\mathbf{z}$. The remaining elements are set to $0$. We reverse the sign of the mapping for $\mathbf{x} \in S - T$.*

(**T2**) **Given**: *universum set* $\quad \mathbf{U} = \begin{bmatrix} (\mathbf{x}_1^*)^T \\ \vdots \\ (\mathbf{x}_m^*)^T \end{bmatrix}_{m \times d}$

**Define**: $\quad \mathbf{G} = \begin{bmatrix} \mathbf{1}_{(L-1)\times 1} & & & -\mathbf{I}_{L-1 \times L-1} \\ 0 & \mathbf{1}_{(L-2)\times 1} & & -\mathbf{I}_{L-2 \times L-2} \\ 0 & 0 & \mathbf{1}_{(L-3)\times 1} & -\mathbf{I}_{L-3 \times L-3} \\ & \cdots & & \ddots \end{bmatrix}_{\frac{L(L-1)}{2} \times L}$

**Obtain**: $\mathbf{V} = (\mathbf{G} \otimes \mathbf{U}) \quad$ *where $\otimes$ is the Kronecker product.*

Due to space constraints, the proof of Theorem 2 is provided in the supplementary material. Theorem 2 provides a framework to analyze the sample complexity for PAC-learnability of $\mathcal{H}_{M-SVM}$ and $\mathcal{H}_{MU-SVM}$. A direct observation from Theorem 2 is that MU-SVM is likely to have

a smaller $d_\mathcal{N}$ and hence a lower sample complexity compared to M-SVM. This is seen from (14), where setting $\gamma = 0 \Rightarrow F(\gamma) = \Lambda^2 \Rightarrow G(\gamma) = [2\Lambda^2 R^2]^2$. Hence we always have, $\kappa \leq 2R^2\Lambda^2$ from (13). This gives $\vartheta_{\mathcal{H}_{MU-SVM}} \leq \vartheta_{\mathcal{H}_{M-SVM}}$. In fact $\vartheta_{\mathcal{H}_{MU-SVM}}$ can be significantly smaller than $\vartheta_{\mathcal{H}_{M-SVM}}$ for a well chosen $\gamma \in \{\gamma \geq 0 \ ; \ G(\gamma) \geq 0\}$, resulting to a much smaller $d_\mathcal{N}$ for MU-SVM compared to M-SVM. The trade-off between $m, \Delta$ and the universum data types further ensures low sample-complexity for MU-SVM.

### 3.3 MU-SVM Implementation

Another desirable property of MU-SVM (7) is that it is solvable through state-of-art M-SVM solvers [33, 34]. For every universum sample $\mathbf{x}_{i'}^*$ we create $L$ artificial samples belonging to all the classes, i.e. $(\mathbf{x}_{i'}^*, y_{i'1}^* = 1), \ldots, (\mathbf{x}_{i'}^*, y_{i'L}^* = L)$ and add them to the training set as shown below,

$$(\mathbf{x}_i, y_i, e_{il}, C_i, \xi_i) = \begin{cases} (\mathbf{x}_i, y_i, e_{il}, C, \xi_i) & i = 1 \ldots n \\ (\mathbf{x}_{i'}^*, y_{i'l}^*, -\Delta e_{i'l}, C^*, \zeta_{i'}) & i = n+1 \ldots n + mL; \ i' = 1 \ldots m; \ l = 1 \ldots L \end{cases} \tag{16}$$

**Proposition 3.** *MU-SVM* (7) *after the transformation* (16) *can be solved in the dual form as,*

$$\max_{\boldsymbol{\alpha}} \quad W(\boldsymbol{\alpha}) = -\frac{1}{2}\sum_{i,j}\sum_l \alpha_{il}\alpha_{jl}K(\mathbf{x}_i, \mathbf{x}_j) - \sum_{i,l}\alpha_{il}e_{il} \tag{17}$$

$$s.t. \quad \sum_l \alpha_{il} = 0; \quad \alpha_{i,l} \leq C_i \ \text{if} \ l = y_i; \ \alpha_{i,l} \leq 0 \ \text{if} \ l \neq y_i; \quad i, j = 1 \ldots n + mL, \ l = 1 \ldots L$$

Note that (17) has similar form as the M-SVM's dual form (see [15, 24]), except that (17) has additional $mL$ constraints for the universum samples. Hence, solving MU-SVM using (17) is same as solving an M-SVM problem (3) with $n + mL$ samples.

### 3.4 Model Selection

The MU-SVM (17) has four tunable parameters: $C, C^*$, kernel parameter, and $\Delta$. Successful application of MU-SVM significantly depends on optimal model selection. In this paper we simplify the model selection using a two-step approach,

**(Step a)** First, perform optimal tuning of the $C$ and kernel parameters for M-SVM (2). This equivalently tunes the parameters specific only to the training samples in the MU-SVM formulation.
**(Step b)** Tune $\Delta$ while keeping $C$ and kernel parameters fixed (from Step a). Also $C^* = \frac{nC}{mL}$ is kept fixed to ensure equal contribution of training and universum samples in MU-SVM (7).

The model parameters in Steps (a) & (b) are typically selected through resampling techniques like, leave one out (l.o.o) or stratified cross-validation approaches [35]. Of these approaches, l.o.o provides an almost unbiased estimate of the test error [36]. However, on the downside it can be computationally prohibitive. In this paper we provide a new span definition for MU-SVM in (19), and modify the technique in [17] to derive a new analytic l.o.o bound for MU-SVM. Other span based l.o.o bounds have been derived for alternative versions of the M-SVM formulation [37]. However they do not apply to the C&S's M-SVM and the MU-SVM formulation proposed in this paper. Next, we show that our proposed bound can be sucessfully used for model selection in Steps (a) & (b) thereby avoiding computationally-prohibitive l.o.o and expensive cross-validation. The necessary definitions are provided next.

**Definition 5.** The **(Leave-One-Out procedure)** with the $t^{th}$ training sample dropped involves solving (17) with an additional constraint $\alpha_{tl} = 0; \quad \forall l$. The obtained l.o.o solution $\boldsymbol{\alpha}^t = [\underbrace{\alpha_{11}^t, \ldots, \alpha_{1L}^t}_{\boldsymbol{\alpha}_1^t}, \ldots, \underbrace{\alpha_{t1}^t = 0, \ldots, \alpha_{tL}^t = 0}_{\boldsymbol{\alpha}_t^t = \mathbf{0}}, \ldots]$ with $t^{th}$ sample prediction $\hat{y}_t = $

$\arg\max_l \sum_i \alpha_{il}^t K(\mathbf{x}_i, \mathbf{x}_t)$ gives the leave-one-out error as: $R_{l.o.o} = \frac{1}{n}\sum_{t=1}^n \mathbb{1}[y_t \neq \hat{y}_t]$

**Definition 6.** Support vectors obtained through solving (17) are categorized as: **Type 1** $SV_1 = \{i \, | 0 < \alpha_{iy_i} < C_i\}$ and **Type 2** $SV_2 = \{i \, | \alpha_{iy_i} = C_i\}$.

The set of all support vectors are represented as, $SV = SV_1 \cup SV_2$. Similarly, the set of support vectors for l.o.o solution is given as $SV^t$. Under Definition 6 we prove the following,

**Theorem 3.** *The leave-one-out error is upper bounded as:*

$$R_{l.o.o} \leq \frac{1}{n}\left[|\{t \in SV_1 \cap \mathcal{T} \, |S_t \, max(\sqrt{2}D, \frac{1}{\sqrt{C}}) \geq 1\}| + |\{t \in SV_2 \cap \mathcal{T}\}|\right] \quad (18)$$

*where, $|\cdot| :=$ Cardinality of a set, and $S_t :=$ Span of a Type 1 support vector $\mathbf{x}_t$ given as,*

$$S_t^2 = \min_{\boldsymbol{\beta}} \sum_{i,j}(\sum_l \beta_{il}\beta_{jl})K(\mathbf{x}_i, \mathbf{x}_j) \qquad \text{(new span definition specific to MU-SVM)} \quad (19)$$

$$s.t. \quad \alpha_{il} - \beta_{il} \leq C_i; \quad \forall\{(i \neq t, l)| \, 0 < \alpha_{il} < C_i; \, l = y_i\}$$
$$\alpha_{il} - \beta_{il} \leq 0; \quad \forall\{(i \neq t, l)| \, \alpha_{il} < 0; \, l \neq y_i\}$$
$$\beta_{il} = 0; \quad \forall i \notin SV_1 - \{t\} \, \forall l = 1\ldots L; \quad \beta_{tl} = \alpha_{tl}; \quad \forall l = 1\ldots L; \quad \sum_l \beta_{il} = 0$$

*and D is the Diameter of the smallest hypersphere containing all training samples.*

Please refer to the supplementary material for proof of Theorem 3. The practical utility of (18) is limited due to the significant computational complexity for solving (19) which results to $\sim O(n + mL)^4$ (worst case) to compute (18). To alleviate this, we derive a computationally attractive alternative under the following assumptions,

**Assumption 1.** : For the MU-SVM solution,

- **(A1)** The sets $SV_1$ and $SV_2$ remain the same during the l.o.o procedure.
- **(A2)** The $SV_1$ support vectors have only two active elements i.e. $\forall \boldsymbol{\alpha}_i \in SV_1 \exists \, k \neq y_i$ s.t. $\alpha_{ik} = -\alpha_{iy_i}$.

**Theorem 4.** *Under Assumption 1 the leave-one-out error is upper bounded as:*

$$R_{l.o.o} \leq \frac{1}{n}|\{t \in SV \cap \mathcal{T} \, | \, S_t^2 \geq \boldsymbol{\alpha}_t^\top \sum_{i\in SV}\sum_l \alpha_{il}K(\mathbf{x}_i, \mathbf{x}_t)\}| \quad (20)$$

*where,* $S_t^2 = \begin{cases} \boldsymbol{\alpha}_t^\top[(\mathbf{H}^{-1})_{\mathbf{tt}}]^{-1}\boldsymbol{\alpha}_t & t \in SV_1 \cap \mathcal{T} \\ \boldsymbol{\alpha}_t^\top[K(\mathbf{x}_t, \mathbf{x}_t) \otimes \mathbf{I}_L - \mathbf{K}_t^T\mathbf{H}^{-1}\mathbf{K}_t]\boldsymbol{\alpha}_t & t \in SV_2 \cap \mathcal{T} \end{cases}$ *; $\mathbf{H} := \begin{bmatrix} \mathbf{K}_{SV_1} \otimes \mathbf{I}_L & \mathbf{A}^\top \\ \mathbf{A} & \mathbf{0} \end{bmatrix}$*

$\mathbf{A} := \mathbf{I}_{|SV_1|} \otimes (\mathbf{1}_L)^\top$ *;* $\mathbf{1}_L = [\underbrace{1\,1\ldots1}_{L \, elements}]$ *;* $\mathbf{K}_{SV_1} :=$ *Kernel matrix of the $SV_1$ support vectors*

$(\mathbf{H}^{-1})_{\mathbf{tt}} :=$ *sub-matrix of* $\mathbf{H}^{-1}$ *for indices* $i = [(t-1)L + 1\ldots tL]$ *;* $\mathbf{K}_t = [(\mathbf{k}_t^T \otimes \mathbf{1}_L) \, \mathbf{0}_{L\times|SV_1|}]^T$ *;* $\mathbf{k}_t = n_{|SV_1|\times1}$ *dim vector where $i^{th}$ element is $K(\mathbf{x}_i, \mathbf{x}_t), \forall \mathbf{x}_i \in SV_1$ ; and $\otimes$ is the Kronecker product.*

Theorem 4 provides a good approximation of the l.o.o error (also confirmed from results in Table 3) even when the Assumption 1 is violated just as in [17]. Further, it provides two major advantages over Theorem 3. First, Eq. (20) is valid for both $SV_1$ & $SV_2$ training support vectors and results in a stricter bound. Second, span computation in theorem 4 requires only one matrix inversion $\mathbf{H}^{-1}$. This results in a significant speed-up to $\sim O(n + mL)^3$ for computing the leave-one-out bound using (20) as compared to $\sim O(n + mL)^4$ in (18).

## 4 Empirical Results

We use three real life datasets discussed next:
German Traffic Sign Recognition Benchmark (GTSRB) [38]: The goal is to identify the traffic signs for the speed-zones '30','70' and '80'. Here, the images are represented by their 1568 histogram of gradient (HOG 1) features. For this data we use three kinds of Universum: **(U1)** Random Averaging (RA) : synthetically created by first selecting a random traffic sign from each class ('30','70' and '80') in the training set and aver-

Table 1: Real-life datasets.

| DATASET | TRAIN / TEST SIZE |
|---------|-------------------|
| GTSRB | 300 / 1500 (100 / 500 PER CLASS) |
| ABCDETC | 600 / 400 (150 / 100 PER CLASS) |
| ISOLET | 500 / 500 (100 / 100 PER CLASS) |

aging them. **(U2)** Others: all other *non*-speed traffic signs. **(U3)** 'Priority road' Sign: an exhaustive

search over several non-speed zone traffic signs showed this universum to provide the best performance (see appendix B.4).

Handwritten characters (ABCDETC) [16]: The goal is to identify handwritten digits '0'-'3' using their 10000 ($100 \times 100$) pixel values. We use the characters other than digits as universum i.e., **(U1)** 'A - Z' uppercase letters, **(U2)** 'a - z' lowercase letters, **(U3)** all other symbols like:- ! ? , . ; : = - + / / ( ) \$ % " @ and **(U4)** Random Averaging (RA) generated as above.

Speech-based Isolated Letter recognition (ISOLET) [39]: This is a speech recognition dataset where 150 subjects pronounced each letter 'a - z' twice. The goal is to identify the spoken letters 'a' - 'e' using the 617 dimensional spectral coefficients, contour, sonorant, presonorant, and post-sonorant features. We use two different types of universum: **(U1)** 'Others' which contains all the other speeched letters i.e. 'f' -'z' and **(U2)** Random Averaging (RA) discussed above.

Due to space constraints and to simplify our analyses in later sections, we used only a subset of the training classes. We see similar results using all the classes (results provided in supplementary material B.1). For the model parameters our initial experiments showed linear parameterization to be optimal for GTSRB; hence only linear kernel has been used for it. For ABCDETC and ISOLET an RBF kernel $K(\mathbf{x}_i, \mathbf{x}_j) = \exp(-\gamma \|\mathbf{x}_i - \mathbf{x}_j\|^2)$ with $\gamma = 2^{-7}$ provided optimal results for M-SVM. For all the experiments model selection is done over the range of parameters $C = [10^{-4}, \ldots, 10^3]$, $C^*/C = \frac{n}{mL}$ and $\Delta = [0, 0.01, 0.05, 0.1]$.

**Effectiveness of the MU-SVM formulation** (7)**:** Table 2 provides the average test error for MU-SVM and several other baseline methods over 10 random training/test partitioning of the data in the proportions shown in Table 1. Model selection within each partition is done using stratified 5 Fold CV [35]. Here, SVM$_{\text{OVA}}$ & SVM$_{\text{OVO}}$ denotes the popular ECOC based multiclass extensions one-vs-all (OVA) and one-vs-one (OVO) using binary SVM [2] as the base classifier. Similarly, U-SVM$_{\text{OVA}}$ & U-SVM$_{\text{OVO}}$ uses binary U-SVM [16] as the base classifier. Owing to space constraints, we only show the results for the best performing universum for all the datasets. Also we fix the number of universum samples to $m = 500$. Additional increase in the universum samples do not provide any significant gains (see appendix B.3 for results). The complete set of results using all the universum types are provided in Appendix B.2. For reproducibility of the results we also provide the typical optimal parameters selected through model selection in Appendix B.2.

Table 2 shows that MU-SVM provides lower test errors compared to all the other baseline methods. Specifically, compared to M-SVM, the performance gains using MU-SVM improve significantly up to $\sim 20 - 25\%$. For sufficiently large universum set size, such significant improvements using MU-SVM depend mostly on the statistical characteristics of the universum data. To better understand these statistical characteristics we adopt the technique of 'histogram of projections' (HOP) originally introduced for binary classification in [40]. Here, different from [40], for a given M-SVM / MU-SVM model we project the training samples onto the decision space of their respective classes i.e. $\forall (\mathbf{x}_i, y_i = k)$ we obtain the projection values as $\mathbf{w}_k^\top \mathbf{x}_i - \max_{l \neq k} \mathbf{w}_l^\top \mathbf{x}_i$. In addition we also project the universum samples onto the decision spaces of all the classes i.e. $\forall (\mathbf{x}_{i'}^*)$; project $\forall k; \mathbf{w}_k^\top \mathbf{x}_{i'}^* - \max_{l \neq k} \mathbf{w}_l^\top \mathbf{x}_{i'}^*$.

Finally we generate the class specific histograms of these projection values. In addition to the histograms, we also generate a frequency plot of the predicted labels for the universum samples using the models. Using this HOP visualization we analyze the effectiveness of the universum U3 for GTSRB dataset (see Fig 3). As seen from Fig. 3, the optimal M-SVM model has high separability for the training samples i.e. most of the training samples lie outside the margin borders (+1). In addition, the universum samples U3 are widely spread about the margin-borders and biased towards the positive side of the decision boundary of the sign '30' (Fig. 3(a)); and hence predominantly gets classified as sign '30'(Fig.3(d)). As seen from Figs 3. (e)-(g), applying the MU-SVM model preserves the separability of the training samples and additionally reduces the spread of the universum samples. Following proposition 1, such a model exhibits higher uncertainty on the universum samples' class membership, and uniformly assigns them over all the classes i.e. signs '30','70' and '80' (see Fig. 3(h)). This shows that, the resulting MU-SVM model has higher contradiction (uncertainty) on the universum samples and hence provides better generalization compared to M-SVM. This behavior is consistently seen for the other datasets and universum choices (provided in the supplementary material - Appendix B.6).

**Model Selection using Theorem 4:** Table 3 provides the average $\pm$ std. dev of time taken (in

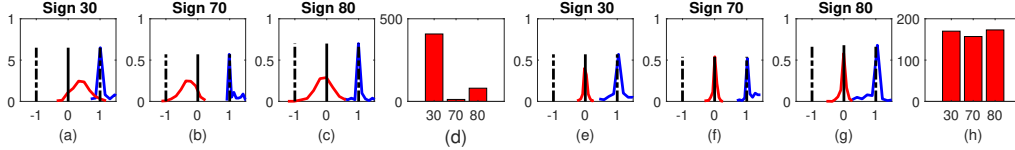

Figure 3: GTSRB: Histograms of projections for training (in blue) and universum **U3** (in red). **M-SVM** ($C = 1$): (a) sign '30'. (b) sign '70'. (c) sign '80'. (d) frequency plot of universum labels. **MU-SVM** ($\Delta = 0$) :(e) sign '30'. (f) sign '70'. (g) sign '80'. (h) frequency plot of universum labels.

Table 2: Mean ($\pm$ standard deviation) of the test errors (in %) over 10 runs of the experimental setting in Table 1. No. of universum samples ($m = 500$).

| DATASET | SVM$_{\text{OVA}}$ | SVM$_{\text{OVO}}$ | M-SVM | U-SVM$_{\text{OVA}}$ | U-SVM$_{\text{OVO}}$ | MU-SVM |
|---|---|---|---|---|---|---|
| GTSRB (USING **U3**) | $7.17 \pm 1.08$ | $7.16 \pm 1.92$ | $7.24 \pm 1.16$ | $6.05 \pm 0.61$ | $5.97 \pm 0.63$ | $\mathbf{5.53 \pm 0.62}$ |
| ABCDETC (USING **U4**) | $28.1 \pm 4.74$ | $29.1 \pm 4.16$ | $27.5 \pm 3.34$ | $26.1 \pm 4.93$ | $26.9 \pm 4.51$ | $\mathbf{22.1 \pm 3.24}$ |
| ISOLET (USING **U2**) | $3.72 \pm 0.6$ | $3.88 \pm 0.44$ | $3.6 \pm 0.31$ | $3.56 \pm 0.55$ | $3.88 \pm 0.63$ | $\mathbf{2.83 \pm 0.32}$ |

seconds) for model selection using Theorem 4 vs. 5-fold CV for 10 runs over the entire range of parameters as well as the respective average test errors. In each experimental run the data is partitioned as in Table 1. We use a desktop with 12 core Intel Xeon @3.5 Ghz and 32 GB RAM. The bound-based model selection is $\sim 2 - 4\times$ faster than 5-fold CV and provides similar test errors. The advantage offered by Theorem 4 is even more pronounced against l.o.o. For instance, comparison with l.o.o for GTSRB dataset showed $\sim 100\times$ improvement in speed using Theorem 4

Table 3: Comparisons for model selection using 5 Fold CV vs. Theorem 4. No. of universum samples ($m = 500$). Model parameters used $C^*/C = \frac{n}{mL}$, $\Delta = [0, 0.01, 0.05, 0.1]$

| | MU$_{\text{SVM}}$ | 5-FOLD CV | | THEOREM 4 | |
|---|---|---|---|---|---|
| | | TEST ERROR (IN %) | TIME ($\times 10^4 sec$) | TEST ERROR (IN %) | TIME ($\times 10^4 sec$) |
| GTSRB | **U1** | $6.9 \pm 0.9$ | $3.1 \pm 0.5$ | $6.9 \pm 0.9$ | $0.8 \pm 0.2$ |
| | **U2** | $7.4 \pm 0.9$ | $3.2 \pm 0.9$ | $7.1 \pm 0.8$ | $0.9 \pm 0.3$ |
| | **U3** | $5.5 \pm 0.6$ | $2.9 \pm 0.3$ | $5.2 \pm 0.4$ | $0.9 \pm 0.1$ |
| ABCD ETC | **U1** | $26.1 \pm 4.0$ | $2.8 \pm 0.1$ | $26.1 \pm 3.7$ | $1.1 \pm 0.1$ |
| | **U2** | $24.2 \pm 3.1$ | $2.8 \pm 0.1$ | $24.4 \pm 3.2$ | $1.3 \pm 0.1$ |
| | **U3** | $23.3 \pm 3.2$ | $2.6 \pm 0.2$ | $24.1 \pm 3.8$ | $0.9 \pm 0.09$ |
| | **U4** | $22.1 \pm 3.2$ | $2.6 \pm 0.1$ | $22.0 \pm 2.8$ | $0.9 \pm 0.1$ |
| ISO LET | **U1** | $3.3 \pm 0.3$ | $4.8 \pm 0.9$ | $3.3 \pm 0.3$ | $2.1 \pm 0.5$ |
| | **U2** | $2.8 \pm 0.3$ | $3.1 \pm 0.6$ | $2.6 \pm 0.3$ | $1.9 \pm 0.7$ |

with similar test accuracies (see Appendix B.5). Additional l.o.o results could not be reported owing to its prohibitively slow speed.

## 5 Conclusions

This paper proposes a new formulation for multiclass SVM (MU-SVM). MU-SVM is shown to incur lower sample-complexity for PAC learnability compared to M-SVM by deriving Natarajan dimension. Further, the proposed MU-SVM embodies several useful mathematical properties amenable for: a) its efficient implementation using existing M-SVM solvers, and b) deriving practical analytic bounds that can perform model selection. We empirically show the effectiveness of the formulation as well as the bound. Insights into the workings of MU-SVM using HOP visualization is also provided.

**Acknowledgments**

We thank the anonymous reviewers for their comments which helped improve the quality of the paper.

## Footnotes

[1]Throughout this paper, we use index $i, j$ for training samples, $i'$ for universum samples and $k, l$ for the class labels.

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
