[Supplementary Material]

# Appendix

## Contents

# A Proofs

The references cited in this document follows the numbering used in the main paper.

## A.1 Proof of Proposition 1

The contradiction on any universum sample $\mathbf{x}^* \in \mathcal{X}_\mathcal{U}^*$ is given as (following (4)),

$$\sup_{h \in \mathcal{H}} \mathbb{E}_{\mathcal{D}_\mathcal{U}}[\mathbb{1}_{\{\bigcap_k h(\mathbf{x}^*) \neq k\}}] = 1 - \inf_{h \in \mathcal{H}} \mathbb{E}_{\mathcal{D}_\mathcal{U}}[\mathbb{1}_{\{\bigcup_k h(\mathbf{x}^*) = k\}}]$$

$$= 1 - \min_{\mathbf{w}_1 \ldots \mathbf{w}_L} \mathbb{E}_{\mathcal{D}_\mathcal{U}}[\mathbb{1}_{\{\bigcup_k (\mathbf{w}_k^\top \mathbf{x}^* - \max_{l \neq k} \mathbf{w}_l^\top \mathbf{x}^* > 0)\}}]$$

$$= 1 - \min_{\mathbf{w}_1 \ldots \mathbf{w}_L} \mathbb{E}_{\mathcal{D}_\mathcal{U}}[\mathbb{1}_{\{\sum_k (\mathbf{w}_k^\top \mathbf{x}^* - \max_{l \neq k} \mathbf{w}_l^\top \mathbf{x}^* > 0)\}}] \qquad (21)$$

The first equality follows from De-Morgan's law. The second equality follows from the M-SVM formulation in eq. (2). The third equality follows from the mutual exclusiveness of the events.

Note that, for maximal contradiction following (21) we need $\mathbb{E}_{\mathcal{D}_\mathcal{U}}[\mathbb{1}_{\{\sum_k (\mathbf{w}_k^\top \mathbf{x}^* - \max_{l \neq k} \mathbf{w}_l^\top \mathbf{x}^* > 0)\}}] = 0$.

This necessitates, $\forall k, \mathbf{w}_k^\top \mathbf{x}^* - \max_{l \neq k} \mathbf{w}_l^\top \mathbf{x}^* \not> 0$. Next, for the case when, $\mathbf{w}_k^\top \mathbf{x}^* - \max_{l \neq k} \mathbf{w}_l^\top \mathbf{x}^* < 0 \Rightarrow$ $\exists p \neq k$ such that $\mathbf{w}_p^\top \mathbf{x}^* - \max_{l \neq p} \mathbf{w}_l^\top \mathbf{x}^* > 0$ resulting $\mathbb{E}_{\mathcal{D}_\mathcal{U}}[\mathbb{1}_{\{\sum_k (\mathbf{w}_k^\top \mathbf{x}^* - \max_{l \neq k} \mathbf{w}_l^\top \mathbf{x}^* > 0)\}}] > 0$. Hence the only possible solution is, $\forall k, \mathbf{w}_k^\top \mathbf{x}^* - \max_{l \neq k} \mathbf{w}_l^\top \mathbf{x}^* = 0 \Rightarrow \forall k; \mathbf{w}_k^\top \mathbf{x}^* - \max_{l = \{1 \ldots L\}} \mathbf{w}_l^\top \mathbf{x}^* = 0 \Rightarrow$ $|\mathbf{w}_k^\top \mathbf{x}^* - \max_{l = 1 \ldots L} \mathbf{w}_l^\top \mathbf{x}^*| = 0; \forall k$ (from symmetry of the transformation (16)) $\qquad \square$

## A.2 Proof of Proposition 2

A similar proof is provided for M-SVM in [15]. Here, we provide the proof for MU-SVM formulation. Formulation (7) for binary classification becomes,

$$\min_{\mathbf{w}_1, \mathbf{w}_2, \boldsymbol{\xi}, \boldsymbol{\zeta}} \quad \frac{1}{2}(\|\mathbf{w}_1\|_2^2 + \|\mathbf{w}_2\|_2^2) + C \sum_{i=1}^n \xi_i + C^* \sum_{i'=1}^m (\zeta_{i'1} + \zeta_{i'2}) \qquad (22)$$

$$s.t. \quad (\mathbf{w}_{y_i} - \mathbf{w}_l)^\top \mathbf{x}_i \geq e_{il} - \xi_i; \quad e_{il} = 1 - \delta_{il}, \quad l = 1, 2$$

$$|(\mathbf{w}_k^\top \mathbf{x}_{i'}^* - \max_{l=1,2} \mathbf{w}_l^\top \mathbf{x}_{i'}^*)| \leq \Delta + \zeta_{i'k}; \; \zeta_{i'k} \geq 0, \quad k = 1, 2$$

$$i = 1 \ldots n, \quad i' = 1 \ldots m, \quad \delta_{il} = \left\{ \begin{array}{ll} 1; & y_i = l \\ 0; & y_i \neq l \end{array} \right.$$

The constraints become,
Training samples ($\forall i = 1 \ldots n$)
For any $\mathbf{x}_i \in$ **class 1** labeled as $y_i = +1$; we have

$$(\mathbf{w}_1 - \mathbf{w}_1)^\top \mathbf{x}_i \geq -\xi_i \quad \Rightarrow \quad \xi_i \geq 0$$
$$(\mathbf{w}_1 - \mathbf{w}_2)^\top \mathbf{x}_i \geq 1 - \xi_i \quad \Rightarrow \quad y_i(\mathbf{w}_1 - \mathbf{w}_2)^\top \mathbf{x}_i \geq 1 - \xi_i$$

Similarly, for any $\mathbf{x}_i \in$ **class 2** labeled as $y_i = -1$; we have,

$$(\mathbf{w}_2 - \mathbf{w}_1)^\top \mathbf{x}_i \geq 1 - \xi_i \quad \Rightarrow \quad y_i(\mathbf{w}_1 - \mathbf{w}_2)^\top \mathbf{x}_i \geq 1 - \xi_i$$
$$(\mathbf{w}_2 - \mathbf{w}_2)^\top \mathbf{x}_i \geq -\xi_i \quad \Rightarrow \quad \xi_i \geq 0$$

Universum samples ($\forall i' = 1 \ldots m$)
For any universum sample $\mathbf{x}_{i'}^*$, WLOG we assume $\mathbf{w}_1 \mathbf{x}_{i'}^* \geq \mathbf{w}_2 \mathbf{x}_{i'}^*$. Then,
When k = 1 we have $|\mathbf{w}_1^\top \mathbf{x}_{i'}^* - \max_{l=1,2} \mathbf{w}_l^\top \mathbf{x}_{i'}^*| \leq \Delta + \zeta_{i'k} \quad \Rightarrow \zeta_{i'k} \geq -\Delta$ (true $\because \zeta_{i'k} \geq 0$).

When k = 2 we have $|\mathbf{w}_2^\top \mathbf{x}_{i'}^* - \max_{l=1,2} \mathbf{w}_l^\top \mathbf{x}_{i'}^*| \leq \Delta + \zeta_{i'k} \quad \Rightarrow |\mathbf{w}_2^\top \mathbf{x}_{i'}^* - \mathbf{w}_1^\top \mathbf{x}_{i'}^*| \leq \Delta + \zeta_{i'k}, \; \zeta_{i'k} \geq 0.$

Hence, eq. (22) can be re-written as,

$$\min_{\mathbf{w}_1,\mathbf{w}_2,\boldsymbol{\xi},\boldsymbol{\zeta}} \quad \frac{1}{2}(\|\mathbf{w}_1\|_2^2 + \|\mathbf{w}_2\|_2^2) + C\sum_{i=1}^{n}\xi_i + C^*\sum_{i'=1}^{m}\zeta_i' \tag{23}$$

$$s.t. \quad y_i(\mathbf{w}_1 - \mathbf{w}_2)^\top \mathbf{x}_i \geq 1 - \xi_i; \quad \xi_i \geq 0, \quad i = 1\ldots n$$
$$|(\mathbf{w}_1 - \mathbf{w}_2)^\top \mathbf{x}_{i'}^*| \leq \Delta + \zeta_{i'}; \quad \zeta_{i'} \geq 0, \quad i' = 1\ldots m$$

The solution to the KKT system of (23) satisfies $\mathbf{w_1} = -\mathbf{w}_2$. Hence replacing $\mathbf{w} = \mathbf{w}_1 - \mathbf{w}_2$ in (23) still solves (22). This is the U-SVM formulation in [16] with $b = 0$. $\qquad\square$

### A.3 Proof of Theorem 1

See [26, 32] for proofs.

### A.4 Proof of Theorem 2

To prove this theorem we first prove the following Lemmas A.1 and A.2.

**Lemma A.1.** *The VC dimension of the hypothesis class,* $\mathcal{B} = \Big\{\mathbf{x} \rightarrow sign(\mathbf{w}^T\mathbf{x}) : \mathbf{x} \in \Re^d;\ y\mathbf{w}^T\mathbf{x} \geq 1\ ;\ \|\mathbf{w}\|^2 \leq \Lambda^2\Big\},$ *assuming* $\|\mathbf{x}\|^2 \leq R^2$ *follows,*

$$VC(\mathcal{B}) \leq min(d+1, R^2\Lambda^2) \tag{24}$$

**Proof**: For the first term i.e. $VC(\mathcal{B}) \leq d + 1$ refer to Example 3.2 in [27]. For the second term i.e. $VC(\mathcal{B}) \leq R^2\Lambda^2$ see [27] Theorem 4.2 . $\qquad\square$

**Lemma A.2.** *The VC dimension of the hypothesis class,* $\mathcal{B} = \Big\{\mathbf{x} \rightarrow sign(\mathbf{w}^T\mathbf{x}) : \mathbf{x} \in \Re^d;\ y\mathbf{w}^T\mathbf{x} \geq 1\ ;\ \|\mathbf{w}\|^2 \leq \Lambda^2\ ;\ |\mathbf{w}^T\mathbf{u}_j| \leq \Delta\ ;\ u_j \in \Re^d\ ;\ \forall j = 1\ldots m\Big\},$ *assuming* $\|\mathbf{x}\|^2 \leq R^2$ *follows,*

$$VC(\mathcal{B}) \leq min(d+1, \kappa) \tag{25}$$

*where,*

$$\kappa \leq \frac{(\Lambda^2 + \gamma m\Delta^2)R^2 + \sqrt{[(\Lambda^2 + \gamma m\Delta^2)R^2]^2 - 4\gamma(\Lambda^2 + \gamma m\Delta^2)tr[(\mathbf{I} + \gamma\mathbf{U}\mathbf{U}^T)^{-1}(\mathbf{U}\mathbf{X}^T\mathbf{X}\mathbf{U}^T)]}}{2};$$
$$with\ \forall \gamma \in\ \{\gamma \geq 0; [(\Lambda^2 + \gamma m\Delta^2)R^2]^2 - 4\gamma(\Lambda^2 + \gamma m\Delta^2)tr[(\mathbf{I} + \gamma\mathbf{U}\mathbf{U}^T)^{-1}(\mathbf{U}\mathbf{X}^T\mathbf{X}\mathbf{U}^T)] \geq 0\}$$

and, $\mathbf{U} = \begin{bmatrix} (\mathbf{u}_1)^T \\ \vdots \\ (\mathbf{u}_m)^T \end{bmatrix}_{m\times d}$ and, $\mathbf{X} = \begin{bmatrix} (\mathbf{x}_1)^T \\ \vdots \\ (\mathbf{x}_{VC(\mathcal{B})})^T \end{bmatrix}_{VC(\mathcal{B})\times d}$ is the training set shattered by $\mathcal{B}$.

**Proof:** For the first term i.e. $VC(\mathcal{B}) \leq d+1$ refer to Example 3.2 in [27]. The proof for the second term $VC(\mathcal{B}) \leq \kappa$ follows next,

$$\mathcal{B} = \Big\{\mathbf{x} \rightarrow sign(\mathbf{w}^T\mathbf{x}) : \mathbf{x} \in \Re^d;\ y\mathbf{w}^T\mathbf{x} \geq 1\ ;\ \|\mathbf{w}\|^2 \leq \Lambda^2\ ;\ |\mathbf{w}^T\mathbf{u}_j| \leq \Delta\ ;\ u_j \in \Re^d\ ;\ \forall j = 1\ldots m\Big\}$$

$$= \Big\{\mathbf{x} \rightarrow sign(\mathbf{w}^T\mathbf{x}) : \mathbf{x} \in \Re^d;\ y\mathbf{w}^T\mathbf{x} \geq 1\ ;\ \|\mathbf{w}\|^2 \leq \Lambda^2\ ;\ \Big\|\Big[|(\mathbf{w}^T\mathbf{u}_j)|_{j=1}^m\Big]\Big\|_\infty \leq \Delta\Big\}$$

$$\subseteq \Big\{\mathbf{x} \rightarrow sign(\mathbf{w}^T\mathbf{x}) : \mathbf{x} \in \Re^d;\ y\mathbf{w}^T\mathbf{x} \geq 1\ ;\ \|\mathbf{w}\|^2 \leq \Lambda^2\ ;\ \mathbf{w}^T(\sum_{j=1}^{m}\mathbf{u}_j\mathbf{u}_j^T)\mathbf{w} \leq m\Delta^2\Big\} := \mathcal{G} \text{ (norm equivalence)}$$

Clearly $d_{\mathcal{B}} = VC(\mathcal{B}) \leq VC(\mathcal{G}) = d_{\mathcal{G}}$. Obviously $\mathbf{X}$ is also shattered by $\mathcal{G}$. This means $\forall \mathbf{x}_i \in \mathbf{X}$ and $\forall y_i \in \{-1, +1\}$; $\exists \mathbf{w} \in \mathcal{G}$ which satisfies $y_i\mathbf{w}^T\mathbf{x}_i \geq 1$. Summing over all the samples in $\mathbf{X}$ gives, $\exists \mathbf{w} \in \mathcal{G}$ which satisfies $\sum_{i=1}^{d_{\mathcal{B}}} y_i\mathbf{w}^T\mathbf{x}_i \geq d_{\mathcal{B}}$. This also implies $\sup_{\mathbf{w}\in\mathcal{G}} \sum_{i=1}^{d_{\mathcal{B}}} y_i\mathbf{w}^T\mathbf{x}_i \geq d_{\mathcal{B}}$.

Now, since the above relation holds for any random selection of $\mathbf{y} \in \{-1, +1\}^{d_{\mathcal{B}}}$. Hence for any $\mathbf{y}$ following the radamacher distribution we have,

$$d_{\mathcal{B}} \leq \mathbb{E}_{\mathbf{y}}\Big[\sup_{\mathbf{w} \in \mathcal{G}} \sum_{i=1}^{d_{\mathcal{B}}} y_i \mathbf{w}^T \mathbf{x}_i\Big]$$

$$= \mathbb{E}_{\mathbf{y}}\left[\sup_{\substack{||\mathbf{w}||^2 \leq \Lambda^2 \\ \mathbf{w}^T \mathbf{U}^T \mathbf{U} \mathbf{w} \leq m\Delta^2}} \mathbf{y}^T \mathbf{X} \mathbf{w}\right]$$

$$\leq \mathbb{E}_{\mathbf{y}}\left[\sup_{||\mathbf{w}||^2 + \gamma(\mathbf{w}^T \mathbf{U}^T \mathbf{U} \mathbf{w}) \leq \Lambda^2 + \gamma m\Delta^2} \mathbf{y}^T \mathbf{X} \mathbf{w}\right] \quad ; \ (\forall \gamma \geq 0 \quad \text{relaxes the constraint})$$

$$= \mathbb{E}_{\mathbf{y}}\left[\sup_{\mathbf{w}^T (\frac{\mathbf{I} + \gamma \mathbf{U}^T \mathbf{U}}{\Lambda^2 + \gamma m\Delta^2}) \mathbf{w} \leq 1} \mathbf{y}^T \mathbf{X} \mathbf{w}\right] \tag{26}$$

$$= \mathbb{E}_{\mathbf{y}}\left[\Big(\frac{\mathbf{I} + \gamma \mathbf{U}^T \mathbf{U}}{\Lambda^2 + \gamma m\Delta^2}\Big)^{-\frac{1}{2}} \mathbf{X}^T \mathbf{y}\right] \quad \text{(the stationary point of the sup problem in (26))}$$

$$\leq \left[\mathbb{E}_{\mathbf{y}}\Big[\Big(\frac{\mathbf{I} + \gamma \mathbf{U}^T \mathbf{U}}{\Lambda^2 + \gamma m\Delta^2}\Big)^{-\frac{1}{2}} \mathbf{X}^T \mathbf{y}\Big]^2\right]^{\frac{1}{2}} \quad \text{(Jensen's inequality)}$$

$$= \left[tr\Big[\mathbf{X}\Big(\frac{\mathbf{I} + \gamma \mathbf{U}^T \mathbf{U}}{\Lambda^2 + \gamma m\Delta^2}\Big)^{-1} \mathbf{X}^T\Big]\right]^{\frac{1}{2}} \quad (\because \mathbf{y} \text{ behaves as radamacher variables})$$

$$\Leftrightarrow d_{\mathcal{B}}^2 \leq \Big(\Lambda^2 + \gamma m\Delta^2\Big) tr\Big[\mathbf{X}(\mathbf{I} + \gamma \mathbf{U}^T \mathbf{U})^{-1} \mathbf{X}^T\Big] \quad (\because d_{\mathcal{B}} > 0)$$

$$= \Big(\Lambda^2 + \gamma m\Delta^2\Big) tr\Big[\mathbf{X}^T \mathbf{X}(\mathbf{I} + \gamma \mathbf{U}^T \mathbf{U})^{-1}\Big]$$

$$= \Big(\Lambda^2 + \gamma m\Delta^2\Big) tr\Big[\mathbf{X}^T \mathbf{X}\Big(\mathbf{I} - \gamma \mathbf{U}^T (\mathbf{I} + \gamma \mathbf{U} \mathbf{U}^T)^{-1} \mathbf{U}\Big)\Big] \quad \text{(Sherman-Morrison-Woodbury formula)}$$

$$= \Big(\Lambda^2 + \gamma m\Delta^2\Big) \left[tr\Big[\mathbf{X}^T \mathbf{X}\Big] - \gamma tr\Big[\mathbf{X} \mathbf{U}^T (\mathbf{I} + \gamma \mathbf{U} \mathbf{U}^T)^{-1} \mathbf{U} \mathbf{X}^T\Big]\right]$$

$$\leq \Big(\Lambda^2 + \gamma m\Delta^2\Big) \left[d_{\mathcal{B}} R^2 - \gamma tr\Big[(\mathbf{I} + \gamma \mathbf{U} \mathbf{U}^T)^{-1} (\mathbf{U} \mathbf{X}^T \mathbf{X} \mathbf{U}^T)\Big]\right]; \quad \text{(from assumption } ||\mathbf{x}||^2 \leq R^2)$$

$$\therefore d_{\mathcal{B}}^2 \leq \Big(\Lambda^2 + \gamma m\Delta^2\Big) d_{\mathcal{B}} R^2 - \gamma\Big(\Lambda^2 + \gamma m\Delta^2\Big) tr\Big[(\mathbf{I} + \gamma \mathbf{U} \mathbf{U}^T)^{-1} (\mathbf{U} \mathbf{X}^T \mathbf{X} \mathbf{U}^T)\Big]; \quad \forall \gamma \geq 0$$

Solving the above quadratic inequality gives, $\Big(d_{\mathcal{B}} - s_1\Big)\Big(d_{\mathcal{B}} - s_2\Big) \leq 0$ where,

$$s_1 = \frac{(\Lambda^2 + \gamma m\Delta^2)R^2 + \sqrt{[(\Lambda^2 + \gamma m\Delta^2)R^2]^2 - 4\gamma(\Lambda^2 + \gamma m\Delta^2)tr[(\mathbf{I} + \gamma \mathbf{U} \mathbf{U}^T)^{-1}(\mathbf{U} \mathbf{X}^T \mathbf{X} \mathbf{U}^T)]}}{2}$$

$$s_2 = \frac{(\Lambda^2 + \gamma m\Delta^2)R^2 - \sqrt{[(\Lambda^2 + \gamma m\Delta^2)R^2]^2 - 4\gamma(\Lambda^2 + \gamma m\Delta^2)tr[(\mathbf{I} + \gamma \mathbf{U} \mathbf{U}^T)^{-1}(\mathbf{U} \mathbf{X}^T \mathbf{X} \mathbf{U}^T)]}}{2}$$

$$\Rightarrow s_2 \leq d_{\mathcal{B}} \leq s_1$$

Hence, $d_{\mathcal{B}} \leq \frac{(\Lambda^2 + \gamma m\Delta^2)R^2 + \sqrt{[(\Lambda^2 + \gamma m\Delta^2)R^2]^2 - 4\gamma(\Lambda^2 + \gamma m\Delta^2)tr[(\mathbf{I} + \gamma \mathbf{U} \mathbf{U}^T)^{-1}(\mathbf{U} \mathbf{X}^T \mathbf{X} \mathbf{U}^T)]}}{2}$; $\forall \gamma \in \{\gamma \geq 0; [(\Lambda^2 + \gamma m\Delta^2)R^2]^2 - 4\gamma(\Lambda^2 + \gamma m\Delta^2)tr[(\mathbf{I} + \gamma \mathbf{U} \mathbf{U}^T)^{-1}(\mathbf{U} \mathbf{X}^T \mathbf{X} \mathbf{U}^T)] \geq 0\}$ (ensuring real-solutions with *discriminant* under the square root $\geq 0$). $\qquad \square$

Finally the proof for theorem 2 follows,

Proof for $\mathcal{H}_{M-SVM}$

The proof follows the notion of binary reduction adopted in [32] Theorem 22. Here however, rather than analyzing a generic class of hyper-planes we analyze the $\mathcal{H}_{M-SVM}$ hypothesis class as shown in (2). First, we flatten the vector-valued functions as $\mathbf{w} = [\mathbf{w}_1, \ldots, \mathbf{w}_L] \in \Re^{dL}$. Next, let $d_{\mathcal{N}}$ be the natarajan dimension of $\mathcal{H}_{M-SVM}$ and $S = \{\mathbf{x}_1, \ldots, \mathbf{x}_{d_{\mathcal{N}}}\}$ be a set that is shattered using the functions $f_1(\mathbf{x}), f_2(\mathbf{x})$ following definition 4. That is, $\exists h_{\mathbf{w}}(\mathbf{x}) = \underset{l=1,\ldots,L}{\mathrm{argmax}} \ \mathbf{w}_l^T \mathbf{x} \in \mathcal{H}_{M-SVM}$ s.t

$\forall \mathbf{x} \in T \subseteq S$ ; $h_{\mathbf{w}}(\mathbf{x}) = f_1(\mathbf{x})$ and $\forall \mathbf{x} \in S - T$ ; $h_{\mathbf{w}}(\mathbf{x}) = f_2(\mathbf{x})$ with $f_1(\mathbf{x}) \neq f_2(\mathbf{x})$.

With this in place, we define the transformation below following [32],

**Define**: a mapping $\phi : \Re^d \to \Re^{dL}$ as ,

$$
\mathbf{z} = \phi(\mathbf{x}) = \begin{cases} (\mathbf{0}_{d\times 1}, \ldots, \underset{f_1(\mathbf{x})=l}{\mathbf{x}}, \cdots \underset{f_2(\mathbf{x})=k}{-\mathbf{x}}, \ldots, \mathbf{0}_{d\times 1})_{dL\times 1}; \ \forall \mathbf{x} \in T \subseteq S \\ (\mathbf{0}_{d\times 1}, \ldots, \underset{f_1(\mathbf{x})=l}{-\mathbf{x}}, \cdots \underset{f_2(\mathbf{x})=k}{\mathbf{x}}, \ldots, \mathbf{0}_{d\times 1})_{dL\times 1}; \ \forall \mathbf{x} \in S - T \end{cases}
$$

**Obtain**: $\quad \mathbf{Z} = \begin{bmatrix} (\mathbf{z}_1)^T \\ \vdots \\ (\mathbf{z}_{d_{\mathcal{N}}})^T \end{bmatrix}.$

Basically, the transformation $\phi$ maps a sample $\mathbf{x} \in \Re^d$ from the shattered set $S$ to a $dL$ - dimension vector $\mathbf{z}$; where for any $\mathbf{x} \in T$ with $f_1(\mathbf{x}) = l$ and $f_2(\mathbf{x}) = k$; we copy the $\mathbf{x}$ vector onto $l(d-1) + 1 \ldots ld$-th position and $-\mathbf{x}$ vector onto $k(d-1)+1 \ldots kd$-th position of $\mathbf{z}$. The remaining elements are set to $0$. We reverse the sign of the mapping for $\mathbf{x} \in S - T$. Under the above transformation we have the following,

- $||\mathbf{z}||^2 = 2||\mathbf{x}||^2 \leq 2R^2; \quad \forall \mathbf{x} \in S$ (from assumption $||\mathbf{x}|| \leq R^2$).

- $T = \left\{ \mathbf{x} \in S \ : \mathbf{w}_{f_1(\mathbf{x})}^T \geq \mathbf{w}_{f_2(\mathbf{x})}^T + 1; \ ||\mathbf{w}||^2 \leq \Lambda^2 \right\} = \left\{ \mathbf{x} \in S \ : \mathbf{w}^T \mathbf{z} \geq 1; ||\mathbf{w}||^2 \leq \Lambda^2 \right\} \subseteq \left\{ \mathbf{z} \to sign(\mathbf{w}^T \mathbf{z}) \ : y\mathbf{w}^T \mathbf{z} \geq 1 \ ; \ ||\mathbf{w}||^2 \leq \Lambda^2 \right\} := \mathcal{B}$ where $y = +1$ ; if $\mathbf{x} \in T$ and $y = -1$ ; if $\mathbf{x} \in S - T$

Hence, for any subset $T \subseteq S$ we can map a binary labeling problem $\mathcal{B}$. This gives, the number of subsets of $S = 2^{d_{\mathcal{N}}} \leq$ number of possible labeling of $\mathcal{B}$ i.e. $O(d_{\mathcal{N}}^{VC(\mathcal{B})})$ (where $VC(\mathcal{B}) := $ VC dimension of $\mathcal{B}$) from Sauer's Lemma. In essence we have, $d_{\mathcal{N}} \leq O(VC(\mathcal{B})log(VC(\mathcal{B})))$ (see [26]'s lemma A.1 ). Finally using Lemma A.1 we have the form in (11).

Proof for $\mathcal{H}_{MU-SVM}$

The proof follows the same argument as above. Here in addition to the transformation of the training data $\mathbf{x} \to \mathbf{z}$ as above we define the following transformation for the given universum set $(\mathbf{x}_j^*)_{j=1}^m$.

**Given**: universum set $\quad \mathbf{U} = \begin{bmatrix} (\mathbf{x}_1^*)^T \\ \vdots \\ (\mathbf{x}_m^*)^T \end{bmatrix}_{m\times d}$

**Define**: $\quad \mathbf{G} = \begin{bmatrix} \mathbf{1}_{(L-1)\times 1} & & & -\mathbf{I}_{L-1\times L-1} \\ 0 & \mathbf{1}_{(L-2)\times 1} & & -\mathbf{I}_{L-2\times L-2} \\ 0 & 0 & \mathbf{1}_{(L-3)\times 1} & -\mathbf{I}_{L-3\times L-3} \\ & \cdots & & \ddots \end{bmatrix}_{\frac{L(L-1)}{2}\times L}$

**Obtain**: $\mathbf{V} = (\mathbf{G} \otimes \mathbf{U})$ and $\mathbf{v}_j^T = (\mathbf{V})_{j^{th}\text{row}}$; where $\otimes$ is the Kronecker product.

Under the above transformation for any subset $T \subseteq S$ as defined in definition 4 for the set $S$ shattered by $\mathcal{H}_{MU-SVM}$ we can map an equivalent binary labeling problem,

$$T = \left\{ \mathbf{x} \in S \; : \mathbf{w}_{f_1(\mathbf{x})}^T \geq \mathbf{w}_{f_2(\mathbf{x})}^T + 1 \; ; \|\mathbf{w}\|^2 \leq \Lambda^2 \; ; \; |\mathbf{w}^T \mathbf{v}_j| \leq \Delta \; ; \; \forall j = 1 \ldots \frac{mL(L-1)}{2} \right\}$$

$$= \left\{ \mathbf{x} \in S \; : \mathbf{w}^T \mathbf{z} \geq 1; \|\mathbf{w}\|^2 \leq \Lambda^2 \; ; \; |\mathbf{w}^T \mathbf{v}_j| \leq \Delta \; ; \; \forall j = 1 \ldots \frac{mL(L-1)}{2} \right\}$$

$$\subseteq \left\{ \mathbf{z} \to sign(\mathbf{w}^T \mathbf{z}) \; : y\mathbf{w}^T \mathbf{z} \geq 1 \; ; \; \|\mathbf{w}\|^2 \leq \Lambda^2 \; ; \; |\mathbf{w}^T \mathbf{v}_j| \leq \Delta \; ; \; \forall j = 1 \ldots \frac{mL(L-1)}{2} \right\} := \mathcal{B}$$

where $y = +1$ ; if $\mathbf{x} \in T$ and $y = -1$ ; if $\mathbf{x} \in S - T$

Similarly as above, $d_{\mathcal{N}} \leq O(VC(\mathcal{B})log(VC(\mathcal{B})))$ (see [26]'s lemma A.1). Using Lemma A.2 we have the final form in (12). $\qquad \square$

## A.5 Proof of Proposition 3

To prove this proposition we first prove the following Lemmas A.3 and A.4.

**Lemma A.3.** *Under transformation* (16)*, the MU-SVM formulation in eq.* (7) *can be solved using,*

$$\min_{\mathbf{w}_1 \ldots \mathbf{w}_L, \xi} \quad \frac{1}{2} \sum_{l=1}^{L} \|\mathbf{w}_l\|_2^2 \quad + \quad \sum_{i=1}^{n+mL} C_i \, \xi_i \qquad (27)$$

$$s.t. \quad (\mathbf{w}_{y_i} - \mathbf{w}_l)^\top \mathbf{x}_i \geq e_{il} - \xi_i \quad i = 1 \ldots n + mL, \quad l = 1 \ldots L$$

**Proof** The contribution due to the universum samples are same for both (7) and (27). For any universum sample $(\mathbf{x}_{i'}^*)$ we identify the active constraints and its overall contribution to the objective function through slack variables i.e.

Equation (7), the overall contribution of the universum sample $\mathbf{x}_{i'}^*$ is,

$$C^* \sum_{k=1}^{L} \zeta_{i'k} \quad s.t. \quad |\mathbf{w}_k^\top \mathbf{x}_{i'}^* - \max_{l=1 \ldots L} \mathbf{w}_l^\top \mathbf{x}_{i'}^*| \leq \Delta + \zeta_{i'k} \quad , \quad \zeta_{i'k} \geq 0, \quad k = 1 \ldots L$$

Case 1: If $k = \underset{l=1 \ldots L}{\operatorname{argmax}} \, \mathbf{w}_l^\top \mathbf{x}_{i'}^*$. The constraint is inactive and $\zeta_{i'k} = 0$.

Case 2: Let $k \neq \underset{l=1 \ldots L}{\operatorname{argmax}} \, \mathbf{w}_l^\top \mathbf{x}_{i'}^*$. Since, $\zeta_{i'k} \geq 0$ the constraint is active if, $-(\mathbf{w}_k^\top \mathbf{x}_{i'}^* - \max_{l \neq k} \mathbf{w}_l^\top \mathbf{x}_{i'}^*) > \Delta$. Then, $\zeta_{i'k} = -[\Delta + (\mathbf{w}_k^\top \mathbf{x}_{i'}^* - \max_{l \neq k} \mathbf{w}_l^\top \mathbf{x}_{i'}^*)]$.

Hence, keeping only the active constraints the overall contribution of the sample $\mathbf{x}_{i'}^*$ is,

$$C^* \sum_{k \in \mathcal{K}_{i'}} -[\Delta + \mathbf{w}_k^\top \mathbf{x}_{i'}^* - \max_{l \neq k} \mathbf{w}_l^\top \mathbf{x}_{i'}^*] \qquad where, \quad \mathcal{K}_{i'} = \{k| - (\mathbf{w}_k^\top \mathbf{x}_{i'}^* - \max_{l \neq k} \mathbf{w}_l^\top \mathbf{x}_{i'}^*) > \Delta\}$$

$$(28)$$

Equation (27), Following eq. (16) for the universum sample $\mathbf{x}_{i'}^*$ we have L artificial samples as $(\mathbf{x}_{i'}^*, y_{i'} = 1), \ldots, (\mathbf{x}_{i'}^*, y_{i'} = L)$ stacked at indices $i = n + (i'-1)L + 1 \ldots n + i'L$. Hence for $\mathbf{x}_{i'}^*$ we have the overall contribution as,

$$C^* \sum_{i=n+(i'-1)L+1}^{n+i'L} \xi_i \qquad s.t. \, (\mathbf{w}_{y_i} - \mathbf{w}_l) \geq -\Delta(1 - \delta_{il}) - \xi_i$$

Now, for $i = n + (i'-1) + k$, we have $\mathbf{x}_i = \mathbf{x}_{i'}^*, y_i = k$. The constraints are,

$(\mathbf{w}_k - \mathbf{w}_1)^\top \mathbf{x}_{i'}^* \geq -\Delta - \xi_i$ $\qquad\qquad\qquad\qquad$ $(\mathbf{w}_k - \mathbf{w}_1)^\top \mathbf{x}_{i'}^* \geq -\Delta - \xi_i$

$\qquad \vdots$ $\qquad\qquad\qquad\qquad\qquad\qquad\qquad\qquad$ $\vdots$

$(\mathbf{w}_k - \mathbf{w}_k)^\top \mathbf{x}_{i'}^* \geq -\xi_i \quad$ (inactive but ensures) $\quad \Rightarrow \qquad\qquad$ $\xi_i \geq 0$

$\qquad \vdots$ $\qquad\qquad\qquad\qquad\qquad\qquad\qquad\qquad$ $\vdots$

$(\mathbf{w}_k - \mathbf{w}_L)^\top \mathbf{x}_{i'}^* \geq -\Delta - \xi_i$ $\qquad\qquad\qquad\qquad$ $(\mathbf{w}_k - \mathbf{w}_L)^\top \mathbf{x}_{i'}^* \geq -\Delta - \xi_i$

This is equivalent to, $-(\mathbf{w}_k^\top \mathbf{x}_{i'}^* - \max_{l \neq k} \mathbf{w}_l^\top \mathbf{x}_{i'}^*) \leq \Delta + \xi_i$. Since, $\xi_i \geq 0$ the constraint is active if, $-(\mathbf{w}_k^\top \mathbf{x}_{i'}^* - \max_{l \neq k} \mathbf{w}_l^\top \mathbf{x}_{i'}^*) > \Delta$, and the contribution becomes, $\xi_i = -[\Delta + \mathbf{w}_k^\top \mathbf{x}_{i'}^* - \max_{l \neq k} \mathbf{w}_l^\top \mathbf{x}_{i'}^*]$. Combining all contributions we get,

$$
\begin{aligned}
C^* \sum_{i=n+(i'-1)L+1}^{n+i'L} \xi_i \quad &\text{s.t. } (\mathbf{w}_{y_i} - \mathbf{w}_l) \geq -\Delta(1 - \delta_{il}) - \xi_i \\
= C^* \sum_{k \in \mathcal{K}_{i'}} -[\Delta + \mathbf{w}_k^\top \mathbf{x}_{i'}^* - \max_{l \neq k} \mathbf{w}_l^\top \mathbf{x}_{i'}^*] \quad &\text{where,} \quad \mathcal{K}_{i'} = \{k| -(\mathbf{w}_k^\top \mathbf{x}_{i'}^* - \max_{l \neq k} \mathbf{w}_l^\top \mathbf{x}_{i'}^*) > \Delta\}
\end{aligned}
\tag{29}
$$

Comparing (28) and (29), the universum sample has similar contribution for both the objective functions in (7) and (27). This is valid for all universum samples. □

**Lemma A.4.** *Eq.* (27) *in dual from can be solved using* (17).

**Proof** This follows from standard KKT system analysis of (27). A similar proof is available in [15, 24]. We reproduce it for completeness and for better readability of the subsequent proofs. The Lagrangian of the (27) is given as,

$$
\text{Lagrangian, } \mathcal{L} = \frac{1}{2} \sum_l \|\mathbf{w}_l\|_2^2 + \sum_{i=1}^{n+mL} C_i \, \xi_i - \sum_{il} \eta_{il}[(\mathbf{w}_{y_i} - \mathbf{w}_l)^T \mathbf{x}_i - e_{il} + \xi_i]
\tag{30}
$$

**KKT System**

$$
\nabla_{\mathbf{w}_l} \mathcal{L} = 0 \qquad \Rightarrow \mathbf{w}_l = \sum_i (C_i \delta_{il} - \eta_{il}) \mathbf{x}_i
\tag{31}
$$

$$
\nabla_{\xi_i} \mathcal{L} = 0 \qquad \Rightarrow \sum_l \eta_{il} = C_i
$$

Complimentary Slackness

$$
\eta_{il}[(\mathbf{w}_{y_i} - \mathbf{w}_l)^T \mathbf{x}_i - e_{il} + \xi_i] = 0 \quad \forall (i,l)
$$

Constraints,

$$
(\mathbf{w}_{y_i} - \mathbf{w}_l)^T \mathbf{x}_i \geq e_{il} + \xi_i \quad \forall (i,l)
$$
$$
\eta_{il} \geq 0
$$

Finally the dual problem is,

$$
\max_{\boldsymbol{\eta}} \quad -\frac{1}{2} \sum_{i,j} \sum_l (C_i \delta_{il} - \eta_{il})(C_j \delta_{jl} - \eta_{jl}) K(\mathbf{x}_i, \mathbf{x}_j) + \sum_{i,l} \eta_{il} e_{il}
\tag{32}
$$

$$
\text{s.t.} \quad \sum_l \eta_{il} = C_i
$$
$$
\eta_{il} \geq 0
$$

Setting $\alpha_{il} = C_i \delta_{il} - \eta_{il}$ we get (17). □

Combining Lemmas A.3 and A.4 we have proposition 3. □

## A.6 Proof of Theorem 3

For this proof we derive the following Lemmas,

- **Lemma A.5** provides some new properties of the solution for MU-SVM dual form in (17).
- **Lemma A.6** : Using these new properties we can follow a similar technique as in [17] to derive a specific condition on Span that holds only for type 1 support vectors contributing to l.o.o error.

**Lemma A.5.** $\forall \boldsymbol{\alpha}_i \in SV_1 = \{i | 0 < \alpha_{il} < C_i; y_i = l\}$,

i. $\sum\limits_k \alpha_{ik}[\sum\limits_j \alpha_{jk} K(\mathbf{x}_i, \mathbf{x}_j) + e_{ik}] = 0$ ; $k = 1 \ldots L$

ii. $\forall k \neq y_i$ with $\alpha_{ik} < 0$ *(strict)*; $\quad \sum\limits_j \alpha_{jk} K(\mathbf{x}_i, \mathbf{x}_j) + e_{ik} = \sum\limits_j \alpha_{jy_i} K(\mathbf{x}_i, \mathbf{x}_j) + e_{iy_i}$ *i.e.*
   *the projection values for the type 1 support vectors for such classes are equal.*

iii. *For any* $\gamma_i \in \{\gamma_i | \sum\limits_k \gamma_{ik} = 0;\ \gamma_{ik} = 0$ *if* $\alpha_i \in SV_1$ *and* $\alpha_{ik} = 0\}$ *we have*
   $\sum\limits_k \gamma_{ik}[\sum\limits_j \alpha_{jk} K(\mathbf{x}_i, \mathbf{x}_j) + e_{ik}] = 0$

**Proof**
For simplicity we provide the proof using linear kernel. The same proof applies for non-linear transformations. The proof uses the KKT system for (27) (see Lemma A.4 eq. (31))

i. $\quad \sum\limits_k \eta_{ik}(\mathbf{w}_{y_i} - \mathbf{w}_k)^T \mathbf{x}_i \quad$ [From (31)]

$$= \sum_k \eta_{ik}(\sum_l \delta_{il}\mathbf{w}_l)^T \mathbf{x}_i - \sum_k \eta_{ik}\mathbf{w}_k^T \mathbf{x}_i$$

$$= \sum_l C_i \delta_{il}\mathbf{w}_l^T \mathbf{x}_i - \sum_k \eta_{ik}\mathbf{w}_k^T \mathbf{x}_i \quad = \quad \sum_k (C_i\delta_{ik} - \eta_{ik})\mathbf{w}_k^T \mathbf{x}_i$$

$$= \sum_k \alpha_{ik} \sum_j \alpha_{jk} K(\mathbf{x}_i, \mathbf{x}_j)$$

From complimentary slackness, if $\alpha_{il} < C_i$ with $y_i = l \Rightarrow \eta_{il} = (C_i\delta_{il} - \alpha_{il}) > 0$. This gives, $(\mathbf{w}_{y_i=l} - \mathbf{w}_{k=l})^T \mathbf{x}_i - e_{ik=l} + \xi_i = 0 \Rightarrow \xi_i = 0$ ( i.e. lies on margin). Now, from complimentary slackness in (31),

$$\sum_k \eta_{ik}[(\mathbf{w}_{y_i} - \mathbf{w}_k)^T \mathbf{x}_i - e_{ik}] = 0$$

$$\Rightarrow \sum_k \alpha_{ik}[\sum_j \alpha_{jk} K(\mathbf{x}_i, \mathbf{x}_j) + e_{ik}] = 0 \quad [\because \eta_{ik}e_{ik} = (C_i\delta_{ik} - \alpha_{ik})e_{ik} = -\alpha_{ik}e_{ik}]$$

ii. From complimentary slackness (31)

$\eta_{ik}[(\mathbf{w}_{y_i} - \mathbf{w}_k)^T \mathbf{x}_i - e_{ik}] = 0 \qquad (\forall k \neq y_i ;\ \alpha_{ik} < 0, \because \xi_i = 0)$

$\Rightarrow (\mathbf{w}_{y_i} - \mathbf{w}_k)^T \mathbf{x}_i - e_{ik} = 0 \qquad (\because \eta_{ik} > 0)$

$\Rightarrow \mathbf{w}_{y_i}^T \mathbf{x}_i = \mathbf{w}_k^T \mathbf{x}_i + e_{ik}$

$\Rightarrow \sum\limits_j \alpha_{jy_i} K(\mathbf{x}_i, \mathbf{x}_j) + e_{iy_i} = \sum\limits_j \alpha_{jk \neq y_i} K(\mathbf{x}_i, \mathbf{x}_j) + e_{ik \neq y_i}$

iii. For any such $\gamma_i$,

$$\sum_k \gamma_{ik}[\sum_j \alpha_{jk} K(\mathbf{x}_i, \mathbf{x}_j) + e_{ik}]$$

$$= \gamma_{iy_i} \sum_j \alpha_{jy_i} K(\mathbf{x}_i, \mathbf{x}_j) + \sum_{k \neq y_i, \alpha_{ik}<0} \gamma_{ik}[\sum_j \alpha_{jk \neq y_i} K(\mathbf{x}_i, \mathbf{x}_j) + e_{ik \neq y_i}]$$

$$= (\gamma_{iy_i} + \sum_{k \neq y_i} \gamma_{iy_i})[\sum_j \alpha_{jy_i} K(\mathbf{x}_i, \mathbf{x}_j)] \quad \text{(from ii above and } \because e_{iy_i} = 1 - \delta_{iy_i} = 0)$$

$$= 0 \quad (\because \sum_k \gamma_{ik} = 0 \text{ by construction})$$

$\square$

With the above properties for the MU-SVM solution from Lemma A.5, we can prove a similar Lemma A.6 as in [17].

**Lemma A.6.** *If in leave-one-out procedure a Type 1 (training) support vector $\mathbf{x}_t \in SV_1 \cap \mathcal{T}$ is recognized incorrectly, then we have,*

$$S_t\, max(\sqrt{2}D, \frac{1}{\sqrt{C}}) > 1$$

*where,*

$$S_t^2 = \min_{\boldsymbol{\beta}} \sum_{i,j} (\sum_l \beta_{il}\beta_{jl})K(\mathbf{x}_i, \mathbf{x}_j)$$

$$
\begin{aligned}
s.t. \quad & \alpha_{il} - \beta_{il} \le C_i; \quad \forall \{(i \ne t, l)|\, \alpha_{il} < C_i;\ l = y_i\} \\
& \alpha_{il} - \beta_{il} \le 0; \quad \forall \{(i \ne t, l)|\, \alpha_{il} > 0;\ l \ne y_i\} \\
& \beta_{il} = 0; \quad \forall (i,l) \notin SV_1 - \{t\} \\
& \beta_{tl} = \alpha_{tl}; \quad \forall l \\
& \sum_l \beta_{il} = 0
\end{aligned}
$$

*$D$ = Diameter of the smallest hypersphere containing all training samples, and $\mathcal{T}$ = Training set*

**Proof**

The *leave-one-out* formulation for MU-SVM with the $t \in \mathcal{T}$ sample dropped is,

$$\max_{\boldsymbol{\alpha}} \quad W(\boldsymbol{\alpha}) = -\frac{1}{2}\sum_{i,j}\sum_l \alpha_{il}\alpha_{jl}K(\mathbf{x}_i, \mathbf{x}_j) - \sum_{i,l}\alpha_{il}e_{il}$$

$$s.t. \quad \sum_l \alpha_{il} = 0 \tag{33}$$

$$\alpha_{il} \le C_i \quad \text{if} \quad l = y_i \quad ; \quad \alpha_{il} \le 0 \quad \text{if} \quad l \ne y_i$$

$$\alpha_{tl} = 0; \quad \forall l \quad \text{(additional constraint)}$$

Then, the *leave-one-out* (l.o.o) error is given as: $R_{l.o.o} = \frac{1}{n}\sum_{t=1}^n \mathbb{1}[y_t \ne \hat{y}_t]$ where, $\boldsymbol{\alpha}^t = [\underbrace{\alpha_{11}^t, \ldots, \alpha_{1L}^t}_{\boldsymbol{\alpha}_1^t}, \ldots, \underbrace{\alpha_{t1}^t = 0, \ldots, \alpha_{tL}^t = 0}_{\boldsymbol{\alpha}_t^t = \mathbf{0}}, \ldots]$ is the solution for (33) and $\hat{y}_t = \arg\max_l \sum_i \alpha_{il}^t K(\mathbf{x}_i, \mathbf{x}_t)$ (estimated class label for the $t^{th}$ sample). The overall proof for the bound on the l.o.o error follows three major steps.

**First**, we construct a feasible solution for (17) using the optimal leave-one-out solution $\boldsymbol{\alpha}^t$. i.e., construct $\boldsymbol{\alpha}^t + \boldsymbol{\gamma}$ as shown below,

$$
\begin{aligned}
& \alpha_{il}^t + \gamma_{il} \le C_i; \quad \forall\, (i,l) \in \{(i,l)|0 < \alpha_{il}^t < C_i;\ l = y_i\} := A_1^t \\
& \alpha_{il}^t + \gamma_{il} \le 0; \quad \forall\, (i,l) \in \{(i,l)|\, \alpha_{il}^t < 0;\ l \ne y_i\} := A_2^t \\
& \gamma_{il} = 0; \quad \forall(i,l) \notin SV_1^t \quad [SV_1^t = A_1^t \cup A_2^t]
\end{aligned}
$$

$$\sum_l \gamma_{il} = 0; \tag{34}$$

Now,

$$I_1 = W(\boldsymbol{\alpha}^t + \boldsymbol{\gamma}) - W(\boldsymbol{\alpha}^t)$$

$$= -\frac{1}{2}\sum_{i,j}\sum_l (\alpha_{il}^t + \gamma_{il})(\alpha_{jl}^t + \gamma_{jl})K(\mathbf{x}_i, \mathbf{x}_j) - \sum_i\sum_l (\alpha_{il}^t + \gamma_{il})e_{il} + \frac{1}{2}\sum_{i,j}\sum_l \alpha_{il}^t\alpha_{jl}^t K(\mathbf{x}_i, \mathbf{x}_j) + \sum_i\sum_l \alpha_{il}^t e_{il}$$

$$= -\frac{1}{2}\sum_{i,j}(\sum_l \gamma_{il}\gamma_{jl})K(\mathbf{x}_i, \mathbf{x}_j) - \sum_{i,j}(\sum_l \gamma_{il}\alpha_{jl}^t)K(\mathbf{x}_i, \mathbf{x}_j) - \sum_i\sum_l \gamma_{il}e_{il}$$

$$= -\frac{1}{2}\sum_{i,j}(\sum_l \gamma_{il}\gamma_{jl})K(\mathbf{x}_i, \mathbf{x}_j) - \sum_{i,l}\gamma_{il}[\sum_j \alpha_{jl}^t K(\mathbf{x}_i, \mathbf{x}_j) + e_{il}]$$

$$= -\frac{1}{2}\sum_{i,j}(\sum_l \gamma_{il}\gamma_{jl})K(\mathbf{x}_i, \mathbf{x}_j) - \sum_l \gamma_{tl}[\sum_j \alpha_{jl}^t K(\mathbf{x}_j, \mathbf{x}_t) + e_{tl}] \quad \text{(Lemma A.5 (iii))} \tag{35}$$

As a special case we set,

$$\boldsymbol{\gamma}_t = [\ldots \underset{y_t}{a}, \ldots, \underset{k^{th}}{-a}, \ldots] = a\mathbf{g}_{y_t k}; \quad (k = \underset{q \neq y_t}{argmax} \sum_j \alpha_{jq}^t K(\mathbf{x}_j, \mathbf{x}_t) \; ; \; \mathbf{g}_{y_t k} = [\ldots \underset{y_t}{1} \ldots \underset{k^{th}}{-1}])$$

Further, we select another $p \in SV_1$ where $\boldsymbol{\gamma}_{p \neq t} = -a\mathbf{g}_{y_t k}$. Finally, we set, $\boldsymbol{\gamma}_i = 0 \; \forall i \notin \{t, p\}$. For such a case,

$$I_1 = -a^2 ||\mathbf{x}_t - \mathbf{x}_p||^2 + a[1 - (\sum_j \alpha_{jy_t}^t K(\mathbf{x}_j, \mathbf{x}_t) - \sum_j \alpha_{jk}^t K(\mathbf{x}_j, \mathbf{x}_t))]$$

$$\geq -\hat{a}^2 D^2 + \hat{a}[1 - (\sum_j \alpha_{jy_t}^t K(\mathbf{x}_j, \mathbf{x}_t) - \sum_j \alpha_{jk}^t K(\mathbf{x}_j, \mathbf{x}_t))] \tag{36}$$

with, $\hat{a} = \frac{1}{2D^2}[1 - (\sum_j \alpha_{jy_t}^t K(\mathbf{x}_j, \mathbf{x}_t) - \sum_j \alpha_{jk}^t K(\mathbf{x}_j, \mathbf{x}_t))]$ (the value that maximizes the R.H.S in (36)) and $D$ = Diameter of the smallest hypersphere containing all training samples.

Now, if; $\quad \hat{a} \leq C \Rightarrow I_1 \geq \frac{1}{4D^2}[1 - (\sum_j \alpha_{jy_t}^t K(\mathbf{x}_j, \mathbf{x}_t) - \sum_j \alpha_{jk}^t K(\mathbf{x}_j, \mathbf{x}_t))] = \frac{1}{2}\hat{a}$

else, $I_1 \geq -C^2 D^2 + C[1 - (\sum_j \alpha_{jy_t}^t K(\mathbf{x}_j, \mathbf{x}_t) - \sum_j \alpha_{jk}^t K(\mathbf{x}_j, \mathbf{x}_t))] = 2CD^2[\hat{a} - \frac{C}{2}] \geq 2CD^2\frac{\hat{a}}{2}$

If there is an error due to leave one out procedure, then $\underset{q \neq y_t}{max} \sum_j \alpha_{jm}^t K(\mathbf{x}_j, \mathbf{x}_t) > \sum_j \alpha_{jy_t}^t K(\mathbf{x}_j, \mathbf{x}_t)$.

This gives, $\quad I_1 > \frac{1}{2}min(C, \frac{1}{2D^2})$ (for l.o.o error) $\tag{37}$

**Second**, we construct a feasible solution for the leave-one-out formulation (33) using the optimal solution for (17). i.e., construct $\boldsymbol{\alpha} - \boldsymbol{\beta}$ as shown below,

$$\alpha_{il} - \beta_{il} \leq C_i; \quad \forall (i, l) \in A_1 - \{t\}; \quad A_1 = \{(i, l)| 0 < \alpha_{il} < C_i; \; l = y_i\}$$
$$\alpha_{il} - \beta_{il} \leq 0; \quad \forall (i, l) \in A_2 - \{t\}; \quad A_2 = \{(i, l)| \alpha_{il} < 0; \; l \neq y_i\}$$
$$\sum_l \beta_{il} = 0;$$
$$\beta_{il} = 0 \qquad \forall (i, l) \notin SV_1 - \{t\} \tag{38}$$
$$\boldsymbol{\beta}_t = \boldsymbol{\alpha}_t$$

with $SV_1 = A_1 \cup A_2 = \{i \,|0 < \alpha_{iy_i} < C_i\}$ such that, it is a feasible solution for (33). As before, define

$$I_2 = W(\boldsymbol{\alpha}) - W(\boldsymbol{\alpha} - \boldsymbol{\beta})$$
$$= -\frac{1}{2}\sum_{i,j}\sum_k \alpha_{il}\alpha_{jl}K(\mathbf{x}_i, \mathbf{x}_j) - \sum_i\sum_l \alpha_{il}e_{il} \quad + \frac{1}{2}\sum_{i,j}\sum_l (\alpha_{il} - \beta_{il})(\alpha_{jl} - \beta_{jl})K(\mathbf{x}_i, \mathbf{x}_j)$$
$$+ \sum_i\sum_l (\alpha_{il} - \beta_{il})e_{il}$$
$$= \frac{1}{2}\sum_{i,j}(\sum_l \beta_{il}\beta_{jl})K(\mathbf{x}_i, \mathbf{x}_j) - \sum_{il}\beta_{il}[\sum_j \alpha_{jl}K(\mathbf{x}_j, \mathbf{x_i}) + e_{il}]$$
$$= \frac{1}{2}\sum_{i,j}(\sum_l \beta_{il}\beta_{jl})K(\mathbf{x}_i, \mathbf{x}_j) \quad \text{(Lemma A.5 (iii))} \tag{39}$$

**Third**, as the final step define,

$$S_t^2 = \min_{\boldsymbol{\beta}} \quad \sum_{i,j} (\sum_l \beta_{il}\beta_{jl}) K(\mathbf{x}_i, \mathbf{x}_j) \tag{40}$$

$$\begin{aligned}
s.t. \quad &\alpha_{il} - \beta_{il} \leq C_i; \quad (i,l) \in A_1 - \{t\} \\
&\alpha_{il} - \beta_{il} \leq 0; \quad (i,l) \in A_2 - \{t\} \\
&\beta_{il} = 0; \quad \forall (i,l) \notin SV_1 - \{t\} \\
&\beta_{tl} = \alpha_{tl}; \quad \forall l \\
&\sum_l \beta_{il} = 0
\end{aligned}$$

Now, let $\boldsymbol{\beta}'$ be the minimizer for (40). For such a $\boldsymbol{\beta}'$

$$\begin{aligned}
I_2(&= \frac{1}{2}S_t^2) \\
&\geq I_1 \quad [\because W(\boldsymbol{\alpha}) \geq W(\boldsymbol{\alpha}+\boldsymbol{\gamma}) \ \forall\boldsymbol{\gamma}; \quad -W(\boldsymbol{\alpha}-\boldsymbol{\beta}) \geq -W(\boldsymbol{\alpha}) \ \forall\boldsymbol{\beta}] \\
&> \frac{1}{2}min(C, \frac{1}{2D^2}) \quad (\text{from}(37))
\end{aligned}$$

$\square$

Finally using Lemma A.6 we analyze the contribution of a sample to the leave-one-out error and make the following arguments,

- **First**, for a sample $(\mathbf{x}_t, y_t)$ which is not a support vector, i.e. $t \notin SV$ and $t \in \mathcal{T}$ (Training set); it lies outside margin borders. Dropping such a sample does not change the original solution of (17). Hence, it does not contribute to an error.

- **Secondly**, for a sample $(\mathbf{x}_t, y_t) \in SV_1 \cap \mathcal{T}$ contributing to leave-one-out error, Lemma A.6 holds i.e. $S_t \, max(\sqrt{2}D, \frac{1}{\sqrt{C}}) > 1$ .

- **Finally**, for a sample $(\mathbf{x}_t, y_t)$ with $t \in SV_2 \cap \mathcal{T}$ we add to the leave-one-out error.

This gives the final form in Theorem 3. $\square$

One observation that follows from Theorem 3 (not discussed in the main text) is,

**Remark 1.** *If the Type 1 training support vectors i.e. $\{t|t \in SV_1 \cap \mathcal{T}\}$ for M-SVM and MU-SVM solutions remain same, then we have $S_t^{SVM} \geq S_t^{MU-SVM}$.*

**Proof** By definition from (19),

$$S_t^2 = \min_{\boldsymbol{\beta}} \quad \sum_{i,j} (\sum_l \beta_{il}\beta_{jl}) K(\mathbf{x}_i, \mathbf{x}_j)$$

$$s.t.$$

$$\boldsymbol{\beta}^{MU-SVM} := \begin{cases} \alpha_{il} - \beta_{il} \leq C_i; & (i,l) \in A_1 - \{t\} \\ \alpha_{il} - \beta_{il} \leq 0; & (i,l) \in A_2 - \{t\} \\ \beta_{il} = 0; & \forall (i,l) \notin SV_1 - \{t\} \\ \beta_{tl} = \alpha_{tl}; & \forall l \\ \sum_l \beta_{il} = 0 \end{cases}$$

If the Type 1 (training) support vectors for M-SVM and MU-SVM solutions remain same, we get the same relation as Lemma A.6 for M-SVM with,

$$\boldsymbol{\beta}^{M-SVM} = \{\beta_{il} \in \boldsymbol{\beta}^{MU-SVM} | \boldsymbol{\beta}_i = \boldsymbol{\alpha}_i ; \forall i \in SV_1 \cap \mathcal{U}\} \quad , \text{where } \mathcal{U} = \text{Universum samples.}$$

i.e. $\boldsymbol{\beta}^{M-SVM} \subseteq \boldsymbol{\beta}^{MU-SVM} \Rightarrow S_t(\boldsymbol{\beta}^{M-SVM}) \geq S_t(\boldsymbol{\beta}^{MU-SVM})$

Loosely speaking, for cases where the type of training support vectors remain same, introducing universum samples through the MU-SVM formulation could result in smaller span values and better

generalization for future test data compared to the M-SVM solution. This observation is in line with our Theorem 2 wherein we show that introducing universum samples through MU-SVM can result to reduced sample complexity and hence improved generalization.

## A.7 Proof of Theorem 4

In this proof we utilize the specific structure of the MU-SVM solution shown in Lemma A.5 and follow a similar analysis as [17]. Here,

- **Lemma A.7** shows that a the span definition can be represented using an equation which solely depends on the MU-SVM solution (17) and its l.o.o solution following definition (5) also shown in (34).
- **Lemma A.8** shows how the compute the span definition in Lemma A.7 through a single matrix inversion.

Finally, with the Lemmas A.7 and A.8 in place we prove Theorem 4.

**Lemma A.7.** *Under **Assumption 1** in Section 3.4 the following equality holds for both Type 1& 2 training support vectors, i.e.* $\forall \mathbf{x}_t \in SV \cap \mathcal{T}$ *we have,*

$$S_t^2 = [\boldsymbol{\alpha}_t^\top \sum_{i \in SV} \sum_l \alpha_{il} K(\mathbf{x}_i, \mathbf{x}_t) - \alpha_{ty_t} \mathbf{g}_{y_t k}^\top \sum_{i \in SV^t} \sum_l \alpha_{il}^t K(\mathbf{x}_i, \mathbf{x}_t)]$$

$$\Rightarrow S_t^2 \geq \sum_l \alpha_{tl} [\sum_{j \in SV} \alpha_{jl} K(\mathbf{x}_j, \mathbf{x}_t)] \quad \text{(during l.o.o procedure)}$$

*with,* $\quad S_t^2 = \{\min_{\boldsymbol{\beta}} \sum_{i,j} (\sum_l \beta_{il}\beta_{jl}) K(\mathbf{x}_i, \mathbf{x}_j) | \boldsymbol{\beta}_t = \boldsymbol{\alpha}_t; \sum_l \beta_{il} = 0 ; (i,j) \in SV_1\} \quad and \quad \mathbf{g}_{y_t k} =$ $[0, \dots \underset{y_t}{1}, \dots, \underset{k^{th}}{-1}, \dots, 0]; \ k = \underset{q \neq y_t}{argmax} \sum_j \alpha_{jq}^t K(\mathbf{x}_j, \mathbf{x}_t)$

**Proof**

Under the Assumption (A1) we set $\boldsymbol{\beta} = \boldsymbol{\gamma} = (\boldsymbol{\alpha} - \boldsymbol{\alpha}^t)$. Then $I_1 = W(\boldsymbol{\alpha}) - W(\boldsymbol{\alpha}^t) = I_2$
A similar analysis as in (35) gives,

$$I_1 = -\frac{1}{2} \sum_{(i,j) \in SV_1} (\sum_l \gamma_{il}\gamma_{jl}) K(\mathbf{x}_i, \mathbf{x}_j) - \sum_l \alpha_{tl} [\sum_{j \in SV} \alpha_{jl}^t K(\mathbf{x}_j, \mathbf{x}_t) + e_{tl}] \tag{41}$$

Note the difference in form compared to (35). This is because now the analysis applies for both type 1& 2 support vectors. Similarly,

$$I_2 = \frac{1}{2} \sum_{i,j} (\sum_l \beta_{il}\beta_{jl}) K(\mathbf{x}_i, \mathbf{x}_j) - \sum_l \alpha_{tl} [\sum_{j \in SV} \alpha_{jl} K(\mathbf{x}_j, \mathbf{x}_t) + e_{tl}] \tag{42}$$

Combining, (41) and (42)

$$\sum_{(i,j) \in SV_1} \sum_l \beta_{il}\beta_{jl} K(\mathbf{x}_i, \mathbf{x}_j) = \sum_l \alpha_{tl} [\sum_{j \in SV} \alpha_{jl} K(\mathbf{x}_j, \mathbf{x}_t) + e_{tl}] - \sum_l \alpha_{tl} [\sum_{j \in SV} \alpha_{jl}^t K(\mathbf{x}_j, \mathbf{x}_t) + e_{tl}]$$
$$\tag{43}$$

Next, let $\boldsymbol{\beta}'$ be the minimizer for (40). Then, $(\boldsymbol{\alpha} - \boldsymbol{\beta}')$ is a feasible solution for (33). Hence,

$$W(\boldsymbol{\alpha}^t) \geq W(\boldsymbol{\alpha} - \boldsymbol{\beta}')$$
$$\Rightarrow W(\boldsymbol{\alpha}) - W(\boldsymbol{\alpha}^t) \leq W(\boldsymbol{\alpha}) - W(\boldsymbol{\alpha} - \boldsymbol{\beta}')$$
$$\Rightarrow \sum_{i,j} (\sum_l \beta_{il}\beta_{jl}) K(\mathbf{x}_i, \mathbf{x}_j) \quad \leq \quad S_t^2$$

However, from Assumption (A1), $\boldsymbol{\beta} = (\boldsymbol{\alpha} - \boldsymbol{\alpha}^t)$ is a feasible solution for (40). Hence for such a $\boldsymbol{\beta}$ we have : $S_t^2 \leq \sum_{i,j}(\sum_l \beta_{il}\beta_{jl})K(\mathbf{x}_i, \mathbf{x}_j)$. Combining the above inequality,

$$S_t^2 = \sum_{i,j}(\sum_l \beta_{il}\beta_{jl})K(\mathbf{x}_i, \mathbf{x}_j) \tag{44}$$

Further, under Assumption (A1) the inequality constraints in (40) are not activated. Hence, $S_t^2 = \{\min_{\boldsymbol{\beta}} \sum_{i,j}(\sum_l \beta_{il}\beta_{jl})K(\mathbf{x}_i, \mathbf{x}_j) | \boldsymbol{\beta}_t = \boldsymbol{\alpha}_t; \sum_l \beta_{il} = 0; (i,j) \in SV_1\}$.

Finally combining (43) and (44) we get,

$$S_t^2 = \sum_l \alpha_{tl}[\sum_{j \in SV} \alpha_{jl}K(\mathbf{x}_j, \mathbf{x}_t) + \cancel{e_{tl}}] - \sum_l \alpha_{tl}[\sum_{j \in SV} \alpha_{jl}^t K(\mathbf{x}_j, \mathbf{x}_t) + \cancel{e_{tl}}] \tag{45}$$

For leave one out error (under Assumption (A2)),

$$-\sum_l \alpha_{tl}[\sum_{j \in SV} \alpha_{jl}^t K(\mathbf{x}_j, \mathbf{x}_t)] = \alpha_{ty_t}[\sum_{j \in SV} \alpha_{jk}^t K(\mathbf{x}_j, \mathbf{x}_t) - \sum_{j \in SV} \alpha_{jy_t}^t K(\mathbf{x}_j, \mathbf{x}_t)]$$

$$\geq 0 \qquad (k = \underset{m \neq y_t}{argmax} \sum_{j \in SV} \alpha_{jm}^t K(\mathbf{x}_j, \mathbf{x}_t))$$

$$\therefore \ S_t^2 \geq \sum_l \alpha_{tl}[\sum_{j \in SV} \alpha_{jl}K(\mathbf{x}_j, \mathbf{x}_t)] \qquad\qquad \square$$

**Lemma A.8.** *The span $S_t^2$ can be efficiently computed as*

$$S_t^2 = \begin{cases} \boldsymbol{\alpha}_t^\top [(\mathbf{H}^{-1})_{\mathbf{tt}}]^{-1}\boldsymbol{\alpha}_t & t \in SV_1 \cap \mathcal{T} \\ \boldsymbol{\alpha}_t^\top [K(\mathbf{x}_t, \mathbf{x}_t) \otimes \mathbf{I}_L - \mathbf{K}_t^T \mathbf{H}^{-1} \mathbf{K}_t]\boldsymbol{\alpha}_t & t \in SV_2 \cap \mathcal{T} \end{cases}$$

*here,* $\quad \mathbf{H} := \begin{bmatrix} \mathbf{K}_{SV_1} \otimes \mathbf{I}_L & \mathbf{A}^\top \\ \mathbf{A} & \mathbf{0} \end{bmatrix}; \quad \mathbf{A} := \mathbf{I}_{|SV_1|} \otimes (\mathbf{1}_L)^\top; \qquad \mathbf{1}_L = [\ \underbrace{1\,1\dots1}_{L\ elements}\ ]$

$(\mathbf{H}^{-1})_{\mathbf{tt}} :=$ *sub-matrix of* $\mathbf{H}^{-1}$ *for indices* $\quad i = (t-1)L + 1 \dots tL$

$\mathbf{K}_{SV_1} \quad := $ *Kernel matrix of Type 1 support vectors.* *and* $\quad \mathbf{K}_t = [(\mathbf{k}_t^T \otimes \mathbf{1}_L)\ \mathbf{0}_{L \times |SV_1|}]^T$

*where,* $\mathbf{k}_t = n_{|SV_1| \times 1}$ *dim vector where $i^{th}$ element is $K(\mathbf{x}_i, \mathbf{x}_t), \forall \mathbf{x}_i \in SV_1$ ; and $\otimes$ is the Kronecker product.*

**Proof**

The span in Lemma A.7 is defined as:

$$S_t^2 = \min_{\boldsymbol{\beta}} \sum_{i,j}(\sum_l \beta_{il}\beta_{jl})K(\mathbf{x}_i, \mathbf{x}_j) \tag{46}$$

$$s.t. \qquad \beta_{tl} = \alpha_{tl} \quad ; \quad \forall l = 1, \dots, L$$

$$\sum_l \beta_{il} = 0 \quad ; \quad \forall (i,j) \in SV_1$$

**Case**($t \in SV_1$)

$$= \min_{\boldsymbol{\beta}} \sum_l (\alpha_{tl}\alpha_{tl})K(\mathbf{x}_t, \mathbf{x}_t) + 2\sum_{i \in SV_1 - \{t\}}\sum_l \alpha_{tl}\beta_{il}K(\mathbf{x}_t, \mathbf{x}_i) + \sum_{(i,j) \in SV_1 - \{t\}}(\sum_l \beta_{il}\beta_{jl})K(\mathbf{x}_i, \mathbf{x}_j)$$

$$s.t. \quad \underbrace{(\mathbf{I}_{|SV_1 - \{t\}|} \otimes \mathbf{1}_L)}_{\mathbf{A}}\boldsymbol{\beta} = \mathbf{0}$$

$$= \min_{\boldsymbol{\beta}} \max_{\boldsymbol{\mu}} \boldsymbol{\alpha}_t^\top [K(\mathbf{x}_t, \mathbf{x}_t) \otimes \mathbf{I}_L]\boldsymbol{\alpha}_t + 2\sum_{i \in SV_1 - \{t\}}\sum_l \alpha_{tl}\beta_{il}K(\mathbf{x}_t, \mathbf{x}_i) + \sum_{(i,j) \in SV_1 - \{t\}}(\sum_l \beta_{il}\beta_{jl})K(\mathbf{x}_i, \mathbf{x}_j)$$

$$+ 2\boldsymbol{\mu}^\top \mathbf{A}\boldsymbol{\beta} + 2\boldsymbol{\alpha}^T \mathbf{A}_{tt}\boldsymbol{\mu} \qquad (\boldsymbol{\mu} := \text{Lagrange Multiplier}, \because \sum_l \alpha_{tl} = 0 \Rightarrow \boldsymbol{\alpha}^T \mathbf{A}_{tt}\boldsymbol{\mu} = 0)$$

$$= \boldsymbol{\alpha}_t^\top [K(\mathbf{x}_t, \mathbf{x}_t) \otimes \mathbf{I}_L]\boldsymbol{\alpha}_t + \min_{\boldsymbol{\beta}} \max_{\boldsymbol{\mu}} \underbrace{2\boldsymbol{\alpha}_t^\top (\mathbf{H}_t^{(-t)})^\top \boldsymbol{\lambda} + \boldsymbol{\lambda}\mathbf{H}^{(-t)}\boldsymbol{\lambda}}_{L(\lambda)} \qquad (\text{with} \quad \boldsymbol{\lambda} = [\boldsymbol{\beta}; \boldsymbol{\mu}])$$

where,  $\mathbf{I}_{|SV_1-\{t\}|} :=$ Identity Matrix of size $|SV_1 - \{t\}|$,
$\mathbf{A}_{tt} :=$  submatrix of $\mathbf{A}$ for indices $(t-1)L+1, \ldots, tL$
$\mathbf{H}^{(-\mathbf{t})} := (t-1)L+1, \ldots, tL$ rows/columns of matrix $\mathbf{H}$ removed; and
$\mathbf{H}_{\mathbf{t}}^{(-\mathbf{t})} := (t-1)L+1, \ldots, tL$ columns of $\mathbf{H}$.
Further, at saddle point : $\triangledown_{\boldsymbol{\lambda}} L(\boldsymbol{\lambda}) = 0 \quad \Rightarrow \boldsymbol{\lambda}^* = -[\mathbf{H}^{(-\mathbf{t})}]^{-1}\mathbf{H}_{\mathbf{t}}^{(-\mathbf{t})}\boldsymbol{\alpha}_t$.
Hence,

$$
\begin{aligned}
S_t^2 &= \boldsymbol{\alpha}_t^\top[(K(\mathbf{x}_t,\mathbf{x}_t) \otimes \mathbf{I}_L) - (\mathbf{H}_{\mathbf{t}}^{(-\mathbf{t})})^\top(\mathbf{H}^{(-\mathbf{t})})^{-1}\mathbf{H}_{\mathbf{t}}^{(-\mathbf{t})}]\boldsymbol{\alpha}_t \\
&= \boldsymbol{\alpha}_t^\top(\mathbf{H}^{-1})_{\mathbf{tt}}\boldsymbol{\alpha}_t
\end{aligned}
\tag{47}
$$

where, $(\mathbf{H}^{-1})_{\mathbf{tt}} \quad :=$ sub-matrix of $\mathbf{H}^{-1}$ for index $i = (t-1)L+1, \ldots, tL$.

**Case** ($t \in SV_2$) A similar analysis as above gives,

$$
S_t^2 = \alpha_t^\top[K(\mathbf{x}_t,\mathbf{x}_t) \otimes \mathbf{I}_L - \mathbf{K}_t^T\mathbf{H}^{-1}\mathbf{K}_t]\boldsymbol{\alpha}_t
\tag{48}
$$

where, $\mathbf{K}_t \quad = [(\mathbf{k}_t^T \otimes \mathbf{1}_L) \quad \mathbf{0}_{L \times |SV_1|}]^T$ and $\mathbf{k}_t = n_{|SV_1| \times 1}$ dim vector where ith element is $K(\mathbf{x}_i,\mathbf{x}_t), \forall \mathbf{x}_i \in SV_1$. $\qquad \square$

Finally, the proof for Theorem 4 has two steps.

– *First*, a sample $(\mathbf{x}_t, y_t)$ which is not a support vector does not contribute to an error.

– *Secondly*, for a sample $(\mathbf{x}_t, y_t)$ with $t \in SV \cap \mathcal{T}$ Lemma A.7 holds. Combining the form of $S_t^2$ in Lemma A.8 completes the proof.

$\qquad \square$

# B  Additional Results

## B.1  M-SVM vs MU-SVM using all training classes

Table 4: Performance comparisons between M-SVM vs. MU-SVM using all training classes.

| DATASETS | # TRAIN / TEST = 700 / 3500 (100 / 500 PER CLASS), # UNIVERSUM (M) = 500 | | | |
|---|---|---|---|---|
| GTSRB | MU-SVM (PRIORITY-ROAD) | MU-SVM (RA) | MU-SVM (NON-SPEED) | - |
| SVM = $11.75 \pm 0.77$ | $9.77 \pm 0.43$ | $11.29 \pm 0.48$ | $11.82 \pm 0.93$ | - |
|  | # TRAIN / TEST = 1500 / 1000 (150 / 100 PER CLASS), # UNIVERSUM (M) = 300 | | | |
| ABCDETC | UPPER | LOWER | SYMBOLS | RA |
| SVM = $42.1 \pm 1.9$ | $41.1 \pm 2.6$ | $40.2 \pm 3.2$ | $39.3 \pm 3.2$ | $38.8 \pm 2.1$ |

Here we show the results using all the classes available in each datasets. We can derive similar conclusions as seen from Table 2

## B.2  Complete Table 2 results using all the universum types.

The complete set of results using all the universum types is reported below. The experiment settings is discussed in Section 4 in the main text.

Table 5: Mean ($\pm$ standard deviation) of the test errors (in %) over 10 runs of the experimental setting in Table 1. No. of universum samples ($m = 500$).

| DATASET | SVM$_{\text{OVA}}$ | SVM$_{\text{OVO}}$ | M-SVM | UNIVERSUM TYPE | U-SVM$_{\text{OVA}}$ | U-SVM$_{\text{OVO}}$ | MU-SVM |
|---|---|---|---|---|---|---|---|
| GTSRB | $7.17 \pm 1.08$ | $7.16 \pm 1.92$ | $7.24 \pm 1.16$ | U1 | $7.18 \pm 0.73$ | $7.23 \pm 1.17$ | $6.98 \pm 0.93$ |
|  |  |  |  | U2 | $6.65 \pm 1.02$ | $6.87 \pm 0.78$ | $7.03 \pm 0.62$ |
|  |  |  |  | U3 | $6.05 \pm 0.61$ | $5.97 \pm 0.63$ | $\mathbf{5.53 \pm 0.62}$ |
| ABCDETC | $28.1 \pm 4.74$ | $29.1 \pm 4.16$ | $27.5 \pm 3.34$ | U1 | $26.4 \pm 4.52$ | $26.2 \pm 3.82$ | $26.1 \pm 3.6$ |
|  |  |  |  | U2 | $25.8 \pm 3.13$ | $27.2 \pm 3.55$ | $24.2 \pm 3.13$ |
|  |  |  |  | U3 | $25.7 \pm 4.09$ | $27.0 \pm 5.59$ | $23.1 \pm 3.23$ |
|  |  |  |  | U4 | $23.7 \pm 4.71$ | $23.9 \pm 4.60$ | $\mathbf{22.1 \pm 3.24}$ |
| ISOLET | $3.72 \pm 0.6$ | $3.88 \pm 0.44$ | $3.6 \pm 0.31$ | U1 | $3.73 \pm 0.7$ | $3.98 \pm 0.9$ | $3.31 \pm 0.27$ |
|  |  |  |  | U2 | $3.56 \pm 0.55$ | $3.88 \pm 0.63$ | $\mathbf{2.83 \pm 0.32}$ |

For reproducibility of the results we also provide the typical parameters obtained through the 5-Fold stratified CV. This is provided in Table 6. All codes shall be made available.

Table 6: Typical optimal parameters for the different methods.

| DATASET | SVM$_{\text{OVA}}$ $(C,\gamma)$ | SVM$_{\text{OVO}}$ $(C,\gamma)$ | M-SVM $(C,\gamma)$ | UNIVERSUM TYPE | U-SVM$_{\text{OVA}}$ $(\frac{C^*}{C},\Delta)$ | U-SVM$_{\text{OVO}}$ $(\frac{C^*}{C},\Delta)$ | MU-SVM $(\frac{C^*}{C},\Delta)$ |
|---|---|---|---|---|---|---|---|
| GTSRB | $0.1 - 1, \times$ | $0.1 - 1, \times$ | $1, \times$ | U1 | $0.2, 0$ | $0.2, 0.01$ | $0.2, 0$ |
|  |  |  |  | U2 | $0.2, 0.1$ | $0.2, 0.1$ | $0.2, 0$ |
|  |  |  |  | U3 | $0.2, 0.1$ | $0.2, 0.01$ | $0.2, 0.1$ |
| ABCDETC | $10, 2^{-7}$ | $1 - 10, 2^{-7}$ | $1, 2^{-7}$ | U1 | $0.3, 0.05 - 0.1$ | $0.3, 0.01$ | $0.3, 0$ |
|  |  |  |  | U2 | $0.3, 0$ | $0.3, 0.01$ | $0.3, 0$ |
|  |  |  |  | U3 | $0.3, 0.05 - 0.1$ | $0.3, 0$ | $0.3, 0$ |
|  |  |  |  | U4 | $0.3, 0$ | $0.3, 0 - 0.05$ | $0.3, 0$ |
| ISOLET | $1 - 10, 2^{-7}$ | $1 - 10, 2^{-7}$ | $1, 2^{-7}$ | U1 | $0.2, 0$ | $0.2, 0$ | $0.2, 0.05$ |
|  |  |  |  | U2 | $0.2, 0$ | $0.2, 0$ | $0.2, 0.01$ |

### B.3 M-SVM vs. MU-SVM using all Universum Types with varying Universum set size

Table 7: Mean ($\pm$ standard deviation) of the test errors (in %) over 10 runs of the experimental setting in Table 1.

| DATA | M-SVM | MU-SVM | NO. OF UNIVERSUM SAMPLES | | |
|------|-------|--------|------|------|------|
| | | | 200 | 500 | 1000 |
| **GTSRB** | $7.24 \pm 1.16$ | U1 | $7.08 \pm 0.71$ | $6.98 \pm 0.93$ | $7.08 \pm 0.43$ |
| | | U2 | $7.23 \pm 0.64$ | $7.03 \pm 0.6$ | $7.01 \pm 0.93$ |
| | | **U3** | $6.97 \pm 1.06$ | $\mathbf{5.53 \pm 0.62}$ | $\mathbf{5.51 \pm 0.78}$ |
| **ABCDETC** | $27.5 \pm 3.3$ | U1 | $26.5 \pm 3.9$ | $26.1 \pm 3.6$ | $26.1 \pm 4.0$ |
| | | U2 | $25 \pm 3.2$ | $24.2 \pm 3.4$ | $24.2 \pm 3.1$ |
| | | U3 | $23.5 \pm 4.3$ | $23.1 \pm 3.2$ | $23.3 \pm 3.2$ |
| | | **U4** | $23.2 \pm 4.8$ | $\mathbf{22.1 \pm 3.2}$ | $\mathbf{22.1 \pm 3.0}$ |
| **ISOLET** | $3.6 \pm 0.3$ | U1 | $3.50 \pm 0.3$ | $3.31 \pm 0.27$ | $3.31 \pm 0.3$ |
| | | **U2** | $3.05 \pm 0.34$ | $\mathbf{2.83 \pm 0.32}$ | $\mathbf{2.82 \pm 0.28}$ |

The table provides comparison between the performance between M-SVM vs. MU-SVM using the different universum types with varying universum set size. As seen from the table, MU-SVM provides better generalization than M-SVM. In fact, for certain universum types, like Priority-Road (U3) for GTSRB, Random Averaging (U4) for ABCDETC and (U2) ISOLET; MU-SVM significantly outperforms the M-SVM model. In such cases, the performance gains improve significantly upto $\sim 20 - 25\%$ with the increase in number of universum samples, and stagnates for a significantly large universum set size. This indicates that for sufficiently large universum set size the effectiveness of MU-SVM depends mostly on the type (statistical characteristics) of the universum data.

### B.4 M-SVM vs. MU-SVM with varying training set size for several universum types using GTSRB dataset

The experiments follow the same setting as in Table 2. However in this case we vary the number of training samples. The universum set size is fixed to $m = 500$ following Table 2 i.e. Further, increase in universum samples does not provide significant performance gains. Table 8 provides the mean and std. deviation of the test errors for the M-SVM and MU-SVM models over 10 random training/test partitioning of the dataset.

Table 8: Mean ($\pm$ standard deviation) of the test errors (in %) over 10 runs for the GTSRB dataset.

| METHODS | NO. OF TRAINING SAMPLES (PER CLASS) | | |
|---|---|---|---|
| | 300 (100) | 750 (250) | 1500 (500) |
| M-SVM | $7.24 \pm 1.16$ | $4.23 \pm 0.49$ | $3.61 \pm 0.38$ |
| (NO PASSING) | $6.98 \pm 0.93$ | $4.64 \pm 0.42$ | $3.49 \pm 0.42$ |
| (NO PASSING FOR TRUCKS) | $6.07 \pm 0.68$ | $4.37 \pm 0.9$ | $3.56 \pm 0.41$ |
| (RIGHT OF WAY) | $6.17 \pm 0.67$ | $4.03 \pm 0.2$ | $3.12 \pm 0.42$ |
| (PRIORITY ROAD) | $5.52 \pm 0.68$ | $3.52 \pm 0.37$ | $3.15 \pm 0.44$ |
| (YIELD RIGHT OF WAY) | $6.2 \pm 0.7$ | $3.83 \pm 0.24$ | $3.11 \pm 0.4$ |
| (STOP) | $6.5 \pm 0.66$ | $4.24 \pm 0.45$ | $3.21 \pm 0.5$ |
| (NO VEHICLES) | $6.24 \pm 0.39$ | $4.29 \pm 0.33$ | $3.16 \pm 0.24$ |
| (NO ENTRY) | $6.17 \pm 0.86$ | $3.95 \pm 0.47$ | $3.31 \pm 0.65$ |
| (DANGER) | $6.01 \pm 0.74$ | $3.92 \pm 0.55$ | $3.49 \pm 0.62$ |
| (SLIPPERY ROAD) | $6.03 \pm 0.64$ | $3.85 \pm 0.28$ | $3.45 \pm 0.62$ |
| RA | $6.98 \pm 0.93$ | $4.12 \pm 0.5$ | $3.44 \pm 0.54$ |
| NON SPEED | $7.03 \pm 0.64$ | $4.32 \pm 0.47$ | $3.65 \pm 0.4$ |

(MU-SVM No. OF UNIVERSUM SAMSPLES = 500)

Figure 9: Typical histogram of projection of training samples ($n = 750$) (shown in blue) and universum samples 'priority-road' ($m = 500$) (shown in red). M-SVM decision functions (with $C = 0.1$) for (a) sign '30'. (b) sign '70'.(c) sign '80'. (d) frequency plot of predicted labels for universum samples using SVM model. MU-SVM decision functions (with $C^*/C = 0.5, \Delta = 0.1$) for (e) sign '30'. (f) sign '70'.(g) sign '80'. (h) frequency plot of predicted labels for universum samples using MU-SVM model.

Figure 10: Typical histogram of projection of training samples ($n = 1500$) (shown in blue) and universum samples 'priority-road' ($m = 500$) (shown in red). M-SVM decision functions (with $C = 0.1$) for (a) sign '30'. (b) sign '70'.(c) sign '80'. (d) frequency plot of predicted labels for universum samples using SVM model. MU-SVM decision functions (with $C^*/C = 1, \Delta = 0.05$) for (e) sign '30'. (f) sign '70'.(g) sign '80'. (h) frequency plot of predicted labels for universum samples using MU-SVM model.

Table 8 shows that MU-SVM with *priority-road* universum provides the best performance. Further, the performance gains due to MU-SVM reduces with the increase in the number of training samples. For further analysis of this result we use the HOP visualization. The histogram of projections for the *priority-road* universum with increased training samples $n = 750, 1500$ are provided in Figs. 9 and 10 respectively. As seen from the figures when the number of training samples is large, the estimation problem becomes well-posed using M-SVM. This is also indicated from the fact that different from Fig 3 now we do not see a huge data-piling effect about the +1 margin borders for the training samples. Such data-piling affect generally happens for ill-posed high-dimensional low sample size settings and has also been previously reported in [4,40] for binary classification problems. For the current setting with well-posed M-SVM solution, application of MU-SVM does not provide a significant improvement over the M-SVM solution. This shows that MU-SVM (similar to binary U-SVM in [4-13,16]) is typically effective for (ill-conditioned) high dimension low sample size settings.

## B.5 Model Selection using Leave-One-Out vs. 5-Fold CV vs. Theorem 4 using GTSRB dataset

We provide additional performance comparisons for model selection using stratified 5-Fold CV and Theorem 4 vs. the leave-one-out procedure. We adopt the same experiment setting as in Table 3.

Table 9: Model selection using leave-one-out (L.O.O) vs. 5 Fold CV vs. Theorem 4. No. of universum samples ($m = 500$). Model parameters used $C^*/C = \frac{n}{mL}$, $\Delta = [0, 0.01, 0.05, 0.1]$. The test error using M-SVM = $7.24 \pm 1.16$.

| MU-SVM | L.O.O | | 5-FOLD CV | | THEOREM 4 | |
|---|---|---|---|---|---|---|
| | TEST ERROR (IN %) | TIME ($\times 10^4 sec$) | TEST ERROR (IN %) | TIME ($\times 10^4 sec$) | TEST ERROR (IN %) | TIME ($\times 10^4 sec$) |
| **U1**(RANDOM AVERAGING) | $6.8 \pm 0.9$ | $186.7 \pm 28.4$ | $6.9 \pm 0.9$ | $3.1 \pm 0.5$ | $6.9 \pm 0.9$ | $0.8 \pm 0.2$ |
| **U2** (OTHERS) | $7.1 \pm 0.9$ | $202.1 \pm 43.9$ | $7.4 \pm 0.9$ | $3.2 \pm 0.9$ | $7.1 \pm 0.8$ | $0.9 \pm 0.3$ |
| **U3** (PRIORITY RD.) | $5.2 \pm 0.6$ | $190.7 \pm 58.7$ | $5.5 \pm 0.6$ | $2.9 \pm 0.3$ | $5.2 \pm 0.4$ | $0.9 \pm 0.1$ |

As seen from Table 9 model selection using Theorem 4 $\sim 100\times$ faster than the leave-one-out procedure and providing similar test errors. Comparisons using the leave-one-out procedure is prohibitively slow and hence could not be reported for the other datasets.

## B.6 Additional Histograms of Projections

This section provides the HOPs for the other universum types for all datasets.

### B.6.1 GTSRB dataset

Figure 11: Typical histogram of projection for training samples ($n = 300$) (shown in blue) and universum samples '*Random Averaging*' ($m = 500$) (shown in red). SVM decision functions (with $C = 1$) for (a) sign '30'. (b) sign '70'.(c) sign '80'. (d) frequency plot of predicted labels for universum samples using SVM model. MU-SVM decision functions (with $C^*/C = 0.2, \Delta = 0$) for (e) sign '30'. (f) sign '70'.(g) sign '80'. (h) frequency plot of predicted labels for universum samples using MU-SVM model.

Figure 12: Typical histogram of projection for training samples ($n = 300$) (shown in blue) and universum samples '*Others*' ($m = 500$) (shown in red). M-SVM decision functions (with $C = 1$) for (a) sign '30'. (b) sign '70'.(c) sign '80'. (d) frequency plot of predicted labels for universum samples using M-SVM model. MU-SVM decision functions (with $C^*/C = 0.2, \Delta = 0.05$) for (e) sign '30'. (f) sign '70'.(g) sign '80'. (h) frequency plot of predicted labels for universum samples using MU-SVM model.

Fig 11 shows the histograms and the frequency plots for M-SVM and MU-SVM models for RA universum. As shown in Fig 11 (a), the M-SVM model already results in a narrow distribution of the universum samples and in turn provides *near* random prediction on the universum samples (Fig. 11(d)). Applying MU-SVM for this case provides no significant change to the M-SVM solution and hence no additional improvement in generalization (also see Table 7 in B.3 and Fig.11 (e)-(h)).

Finally, we provide the histograms and the frequency plots for M-SVM and MU-SVM models for the *Others* Universum samples. In this case, although the universum samples are widely spread about the M-SVM margin-borders (Figs 12(a)-(c)), yet the uncertainity on the universum samples' class membership is uniform across

all the classes (Fig 12(d)). Applying MU-SVM reduces the spread of the universum samples (Figs. 12(e) - (g)). However, it does not significantly increase the contradiction (uncertainity) on the universum samples (compare Figs. 12 (d) vs. (h)). Hence, applying MU-SVM does not provide any significant improvement over the M-SVM model (see Table 7 in B.3).

### B.6.2 ABCDETC Dataset

Figure 13: Typical histogram of projection of training samples ($n = 600$) (shown in blue) and universum samples '*upper case*' letters ($m = 1000$) (shown in red). SVM decision functions (with $C = 1, \gamma = 2^{-7}$) for (a) digit '0'. (b) digit '1'.(c) digit '2'. (d) digit '3'. (e) frequency plot of predicted labels for universum samples using SVM model. MU-SVM decision functions (with $C^*/C = 0.15, \Delta = 0$) for (f) digit '0'. (g) digit '1'.(h) digit '2'. (i) digit '3'.(j) frequency plot of predicted labels for universum samples using MU-SVM model.

Figure 14: Typical histogram of projection of training samples ($n = 600$) (shown in blue) and universum samples '*lower case*' letters ($m = 1000$) (shown in red). SVM decision functions (with $C = 1, \gamma = 2^{-7}$) for (a) digit '0'. (b) digit '1'.(c) digit '2'. (d) digit '3'. (e) frequency plot of predicted labels for universum samples using SVM model. MU-SVM decision functions (with $C^*/C = 0.15, \Delta = 0$) for (f) digit '0'. (g) digit '1'.(h) digit '2'. (i) digit '3'.(j) frequency plot of predicted labels for universum samples using MU-SVM model.

As seen from Figs 13 - 16,

Figure 15: Typical histogram of projection of training samples ($n = 600$) (shown in blue) and universum samples '*symbols*' ($m = 1000$) (shown in red). SVM decision functions (with $C = 1, \gamma = 2^{-7}$) for (a) digit '0'. (b) digit '1'.(c) digit '2'. (d) digit '3'. (e) frequency plot of predicted labels for universum samples using SVM model. MU-SVM decision functions (with $C^*/C = 0.15, \Delta = 0$) for (f) digit '0'. (g) digit '1'.(h) digit '2'. (i) digit '3'.(j) frequency plot of predicted labels for universum samples using MU-SVM model.

Figure 16: Typical histogram of projection of training samples ($n = 600$) (shown in blue) and universum samples '*random averaging*' (RA) ($m = 1000$) (shown in red). SVM decision functions (with $C = 1, \gamma = 2^{-7}$) for (a) digit '0'. (b) digit '1'.(c) digit '2'. (d) digit '3'. (e) frequency plot of predicted labels for universum samples using SVM model. MU-SVM decision functions (with $C^*/C = 0.15, \Delta = 0$) for (f) digit '0'. (g) digit '1'.(h) digit '2'. (i) digit '3'.(j) frequency plot of predicted labels for universum samples using MU-SVM model.

- *Upper* : the M-SVM model results in a narrow distribution of the universum samples and in turn provides *near* random prediction on the universum samples. Applying MU-SVM for this case provides no significant change to multiclass SVM solution and hence no additional improvement in generalization (see Table 7).

- *Lower* : the M-SVM model results in a relatively wider distribution of the universum samples (compared to *Upper*). Applying MU-SVM for this case provides some improvement to the M-SVM (see Table 7).

- *Symbol* and *RA* : the SVM model results in a wide distribution of the universum samples. Further, in both the cases the universum samples are mostly predicted as digit '1'. Applying MU-SVM for this case results to a narrow distribution of the universum samples and increases the uncertainity on the universum samples. This results to a significant improvement to the M-SVM solution (see Table 7 in B.3).

### B.6.3 ISOLET Dataset

Figure 17: Typical histogram of projection of training samples ($n = 500$) (shown in blue) and universum samples '*Others*' ($m = 1000$) (shown in red). SVM decision functions (with $C = 1, \gamma = 2^{-7}$) for (a) letter 'a'. (b) letter 'b'.(c) letter 'c'. (d) letter 'd'. (e) letter 'e'. (f) frequency plot of predicted labels for universum samples using SVM model. MU-SVM decision functions (with $C^*/C = 0.1, \Delta = 0.05$) for (g) letter 'a'. (h) letter 'b'.(i) letter 'c'. (j) letter 'd'. (k) letter 'e'. (l) frequency plot of predicted labels for universum samples using MU-SVM model.

Figure 18: Typical histogram of projection of training samples ($n = 500$) (shown in blue) and universum samples '*RA*' ($m = 1000$) (shown in red). SVM decision functions (with $C = 1, \gamma = 2^{-7}$) for (a) letter 'a'. (b) letter 'b'.(c) letter 'c'. (d) letter 'd'. (e) letter 'e'. (f) frequency plot of predicted labels for universum samples using SVM model. MU-SVM decision functions (with $C^*/C = 0.1, \Delta = 0.1$) for (g) letter 'a'. (h) letter 'b'.(i) letter 'c'. (j) letter 'd'. (k) letter 'e'. (l) frequency plot of predicted labels for universum samples using MU-SVM model.

As seen from Figs 17-18,

- *Others* : the M-SVM model results in a *near* random prediction on the universum samples. Applying MU-SVM for this case reduces the projection of the universum samples but does not result to a significant increase in the uncertaininty of the universum samples, and hence no additional improvement in generalization (see Table 7 in B.3).

- *RA* : the M-SVM model results in a wide distribution of the universum samples. Further, the universum samples are mostly predicted as letter 'd'. Applying MU-SVM for this case results to a narrow distribution of the universum samples and increases the uncertaininty on the universum samples. This results to a significant improvement to the multiclass SVM solution (see Table 7 in B.3).