[Reviews · NeurIPS 2019]

Reviewer 1



Originality : The proposed method is a novel combination of existing methods. This combination comes with theoretical guarantees and practical tools to deal with the obvious tuning complexity. Quality : All claims and proposition are justified and detailed (if not in the paper, in the supplementary material). I did not find flaws in it. The interest of incorporating universum examples is shown (already done in binary case) and the motivation to adapt the framework to multiclass case, is established : forcing a universum example to be neutral for all classes makes sense. The paper proposes a complete piece of work concerning the proposed algorithm. However I'm less convinced by the experimental evaluation : the bibliography points previous works on universum prescription (at least) that are not taken into account in the experimental part. Experiments only compare the proposed method to the ones it is build on (SVM and binary U-SVM), which is not enough : it's a good start but not a complete study. I could even suspect that the comparison could be in favor of universum prespription, surely in terms of complexity, and maybe in terms of accuracy too... an argument in favour of MU-SVM could be the fact that it can easily be applied to small dataset, but avoiding the comparison is not a good option. Clarity : The paper is clearly written and quite easy to follow. There are a few missing notations and some figures don't have a sufficient definition and are ugly once printed. Significance : this is the weak point of the paper. MU-SVM does not apply to large dataset (ok could be a positive argument in some contexts) but even for small datasets it has many hyper-parameters and a high training complexity. Obviously the authors have worked hard on alleviating this problem but I'm not sure it will be enough in practice. Details in the order of appearance in the manuscript: eq(2) : \Lambda not defined eq(3) : \delta_{ij} deifned only for eq(7) l110 : "thet the" eq(11) : d? l143 : I did not catch z_i definition, what are zeros ? sec 3.4 : are we sure that good parameters for M-SVM are good for MU-SVM? l 194 : assumptions are very restrictive... (but experiments semmes to validate the approximation) l 201 the Kronecker definition should have appear earlier in the paper (l 145)

Reviewer 2



AFTER REBUTTAL This is an interesting problem setting. The paper covers a large amount of highly technical material and some proofs contain interesting original ideas. Furthermore, many experiments were run and I have the rare feeling that contrary to many submissions, the results were truly honestly reported rather than cherry picked. Overall, it is clear that the authors worked very hard on this paper and are mathematically competent. However, more work is needed to make the paper meet NeurIPS's standards in terms of exposition: the paper is sloppily written, both in terms of grammar and in terms of mathematical accuracy. Most proofs and mathematical statements are littered with typos and the formulae are often awkward. Although I believe the proofs to be correct (in the sense that the flaws are not fatal) and it is in most cases possible to understand the proof after thinking about it for a long time, the mistakes severely impede understanding. The significance of the results in the experiments section does not appear to be very clearly explained either. In conclusion, the topic and the amount of material covered are amply enough for an excellent submission. However, the work would definitely benefit from substantial revision. Taking everything into account, this is a borderline submission. I am also concerned that the authors did not seem to acknowledge the need for revision in the rebuttal phase. For the good of the community, I hope they will consider making improvements for the camera-ready version.

Reviewer 3



This article deals with multi-category pattern classification in the framework of universum learning. The authors introduce a new multi-class support vector machine which is an extension of the model of Crammer and Singer. For this machine, they establish a bound on the generalization performance and a model selection algorithm based on an extension of the span bound. Empirical results are provided. They are obtained on three well-known data sets. This contribution introduces a relevant solution to the problem addressed. It seems to be technically sound, although I could not check all the proofs in details. Regarding the originality, it is noteworthy that other multi-class extensions of the span bound can be found in the literature, for instance in the PhD manuscript of R. Bonidal. The major choices made should be better justified. For instance, the M-SVM of Crammer and Singer is not the first option that comes to mind since its loss function is not Fisher consistent. The choice of the capacity measure (Natarajan dimension) is also unusual. Indeed, one would have expected to find a scale-sensitive measure, let it be a Rademacher complexity, a metric entropy or a scale-sensitive combinatorial dimension (fat-shattering dimension of the class of margin functions...). Some minor corrections should be made. For instance, The notation for the Kronecker product is used at line 145 but introduced at line 200. Below is a non exhaustive list of typos. Line 42: natarajan -> Natarajan Line 47: propostion -> proposition Line 110: the the -> the Line 247: Table 2 show -> Table 2 shows Line 249: upto -> up to *** Update *** I appreciated the answers to my comments. RĂ©mi Bonidal's PhD manuscript can be found at the following address: http://docnum.univ-lorraine.fr/public/DDOC_T_2013_0066_BONIDAL.pdf I think that the paper of Maximov and Reshetova is not technically sound (to say the least).

[Author Response · NeurIPS 2019]

We thank the reviewers for their inputs and comments. We are providing our responses below.

**[R1]: Practical Utility compared to Universum Prescription:** Computation-wise: a) The training complexity of MU-SVM is not worse than Universum Prescription (UP). In fact, training MU-SVM in primal space is the same as training a single layered network architecture and is faster than UP. In addition, Proposition 3 enables utilizing state-of-art M-SVM solvers [1, 2] for solving MU-SVM. This provides a huge computation advantage for training MU-SVM. b) We agree that concurrency will further improve MU-SVM's training speed and hence its practical utility. All existing parallel/distributed techniques used for UP also applies for MU-SVM in its primal form. Distributed implementation for the dual form of MU-SVM can be achieved through eq. 8 in [3]. This is an ongoing effort. c) Finally, Theorem 4's bound based model selection provides massive speed up for hyperparameter tuning (Table 3, Table 9 Appendix). This significantly reduces the computation complexity of the overall model building process and presents MU-SVM as a highly practical solution.

Accuracy-wise: UP is not designed for high dimension low sample size settings (HDLSS). The hypothesis class induced by the UP architecture is complex and prone to high estimation error for HDLSS. This is also confirmed in our preliminary results (can be added if needed). Base-lining UP in HDLSS settings would be an unfair depiction of UP.

**[R1]: MU-SVM hyperparameters for HDLSS data :-** As rightly pointed out by R1, selecting good hyperparameters for MU-SVM is of utmost importance for its effectiveness in HDLSS settings. We have proposed our solution to alleviate this in Section 3.4. Our empirical results (in Table 2 and 3) confirms that the approach works providing > 20% improvement in test accuracies for MU-SVM compared to M-SVM. Additional results and the selected model parameters for reproducibility of the results are provided in Appendix B2. All our codes will be made public.

**[R2]:** While we concede that due to the space constraints we had to make difficult stylistic choices, these are in line with the universum literature and the associated computational learning theory presentations. The equations spanning multiple lines is unfortunately an artifact of the associated math and space constraints which we will further attempt to fix in the final version. We will address the stylistic elements, as well as some typos that the reviewer suggested. However, we have double checked and maintain that the mathematical correctness of the presented content is accurate. We also appreciate the reviewer's observation about the honest and accurate reporting of the experimental results. We have indeed tried to present a principled evaluation of the proposed approach. We address the specific concerns below:-

**Theorem 2 Typos :** a) in definition of $z_i$, $f_1$ has to be replaced by $f_2$, b) The bound in Appendix (pg iii) is due to Jensen's and not Kahane Khintchine inequality, c) $\mathbf{v}_j^T = (\mathbf{V})_{j^{th}\text{row}}$ is column vector in pg. iv. These will be corrected.

**Theorem 2 additional analysis:** Owing to space constraints we had to choose between a more detailed analysis of Theorem 2 vs. the analytic l.o.o bound for model selection. We believe the later is more important for the practical utility of MU-SVM. As also identified in R1's comments, optimal tuning of the MU-SVM hyperparameters is of utmost necessity for its successful application under HDLSS settings. Table 3 shows that the proposed bound in Theorem 4 provides similar test accuracies, with significant computational gains compared to standard resampling techniques.

**Other Comments: (a)** Bijection between i,i',L in line 158 is defined in eq. 16. **(b)**. We believe the reviewer meant the condition $y_i\mathbf{w}^T\mathbf{x}_i \geq 1$ in page ii. Both these sets $\mathcal{B}$ and $\mathcal{G}$ have this constraint by definition. Statement in pg ii means :

$\exists \mathbf{w} \in \mathcal{G}$ such that $y_i\mathbf{w}^T\mathbf{x}_i \geq 1$. Hence, $d_{\mathcal{B}} \leq \sum\limits_{i=1}^{d_{\mathcal{B}}} y_i\mathbf{w}^T\mathbf{x}_i$. **(c)** The arguments at the top of page iv looks correct.

**[R3]: Multiclass extensions of the span bound:** We were unable to find Remi Bonidal's PhD Manuscript. Based on his DBLP record we believe his work rather focused on model selection using path solution for L2-SVM and not Span bounds. To the best of our knowledge there aren't any Span Bounds available for M-SVM/MU-SVM algorithms. We'd appreciate if the reviewer can point us to the sources and we will appropriately update our introduction section.

**[R3]: Justification of choices**: C & S M-SVM is probably the most popular multiclass SVM approach. Although its not Fisher consistent, this property may have little impact on M-SVMs performance for limited data settings. In fact, the results in [4] (Table 1) shows that M-SVM enjoys the smallest estimation error and (except one-vs-one) the smallest approximation error compared to other popular multiclass approaches. We will add a note on our choice of M-SVM. Natarajan Dimension allows us to extend Fundamental Learning Theorem to multiclass problems (Theorem 1) similar to binary problems using VC dimension. Also it's the most widely researched capacity measure for multiclass SVMs. We agree that there are other improved capacity measures [5,6], however we do not foresee any additional understanding compared to Theorem 2 using such methods. A note on the capacity measures and our choice will be added.

All the other comments are minor text edits and shall be included in an edited version.

[1] F. Lauer and Y. Guermeur, "Msvmpack: a multi-class support vector machine package," *JMLR*, vol. 12, no. Jul, 2011.

[2] U. Dogan *et al.*, "Fast training of multi-class support vector machines," *Rapport technique, University of Copenhagen*, 2011.

[3] N. Parikh and S. Boyd, "Block splitting for distributed optimization," *Math. Program. Comput*, vol. 6, no. 1, 2014.

[4] A. Daniely *et al.*, "Multiclass learning approaches: A theoretical comparison with implications," in *NIPS*, 2012.

[5] Y. Guermeur, "Vc theory of large margin multi-category classifiers," *JMLR*, vol. 8, no. Nov, 2007.

[6] Y. Maximov and D. Reshetova, "Tight risk bounds for multi-class margin classifiers," *arXiv e-prints*, p. arXiv:1507.03040, 2015.


[Meta-Review · NeurIPS 2019]

All the reviewers tend to agree that the paper provides solid technical contributions in the area of universum learning for multi-class SVMs. I am happy to recommend that it is accepted for publication at NeurIPS2019. Nevertheless, there is a great concern on the quality of writing. I therefore urge the authors to keep improving the quality of writing (in terms of both grammar and mathematical accuracy) of the camera-ready version. While the supplementary material is less important than the main paper, the clarity of the proofs presented there play an important role for unfamiliar readers in understanding the major contributions of this work. Hence, I also strongly recommend that the authors keep polishing the supplementary material.